# Towards Understanding Distributional Reinforcement Learning: Regularization, Optimization, Acceleration and Sinkhorn Algorithm

## ABSTRACT

Distributional reinforcement learning (RL) is a class of state-of-the-art algorithms that estimate the whole distribution of the total return rather than only its expectation. Despite the remarkable performance of distributional RL, a theoretical understanding of its advantages over expectation-based RL remains elusive. In this paper, we interpret distributional RL as entropy-regularized maximum likelihood estimation in the *neural Z-fitted iteration* framework, and establish the connection of the resulting risk-aware regularization with maximum entropy RL. In addition, We shed light on the stability-promoting distributional loss with desirable smoothness properties in distributional RL, which can yield stable optimization and guaranteed generalization. We also analyze the acceleration behavior while optimizing distributional RL algorithms and show that an appropriate approximation to the true target distribution can speed up the convergence. Finally, we propose a class of *Sinkhorn distributional RL* algorithm that interpolates between the Wasserstein distance and maximum mean discrepancy (MMD). Experiments on a suite of Atari games reveal the competitive performance of our algorithm relative to existing state-of-the-art distributional RL algorithms.

## 1 INTRODUCTION

The intrinsic characteristics of classical reinforcement learning (RL) algorithms, such as temporal-difference (TD) learning (Sutton & Barto, 2018) and Q-learning (Watkins & Dayan, 1992), are based on the expectation of discounted cumulative rewards that an agent observes while interacting with the environment. In stark contrast to the classical expectation-based RL, a new branch of algorithms called distributional RL estimates the full distribution of total returns and has demonstrated state-of-the-art performance in a wide range of environments (Bellemare et al., 2017a; Dabney et al., 2018b;b; Yang et al., 2019; Zhou et al., 2020; Nguyen et al., 2020). Meanwhile, distributional RL has also enjoyed further benefits in risk-sensitive control, policy exploration settings (Mavrin et al., 2019; Rowland et al., 2019) and robsutness (Sun et al., 2021).

Despite the existence of numerous algorithmic variants of distributional RL with remarkable empirical success, theoretical studies of advantages of distributional RL over expectation-based RL are less established. Existing works include (Lyle et al., 2019) that proved in many tabular and linear approximation settings, distributional RL behaves exactly the same as expectation-based RL. Lyle et al. (2021) investigated the impact of distributional RL from the perspective of representation dynamics. Martin et al. (2020) recently mapped distributional RL problems to a Wasserstein gradient flow problem, treating the distributional Bellman residual as a potential energy functional. Offline distributional RL (Ma et al., 2021) has also been proposed to investigate the efficacy of distributional RL in both risk-neutral and risk-averse domains. Recent works have tended towards closing the gap between theory and practice in distributional RL.

From an algorithmic perspective, the Sinkhorn loss (Sinkhorn, 1967) can be used to tractably approximate the Wasserstein distance and has been successfully applied in numerous crucial machine learning developments, including the Sinkhorn-GAN (Genevay et al., 2018) and Sinkhorn-based adversarial training (Wong et al., 2019). Inspired by the distributional RL literature, Martin et al.

(2020) argues for the use of a second-order stochastic dominance relation to select among a multiplicity of competing solutions via Sinkhorn iteration (Sinkhorn, 1967), which can be useful to manage stochastic uncertainty in RL paradigms. However, a Sinkhorn-based distributional RL algorithm has not yet to be formally proposed and investigated.

In this paper, we theoretically illuminate the superiority of distributional RL over expectation-based RL from the perspectives of regularization, optimization, acceleration, and representation. Specifically, we simplify distributional RL as a *neural Z-fitted iteration*, within which we establish an equivalence between distributional RL and a form of entropy-regularized maximum likelihood estimation (MLE). We also demonstrate that the resulting novel cross entropy regularization correlates strongly with the behavior of maximum entropy RL. By incorporating a histogram distributional loss, we further achieve stable optimization and guaranteed generalization of distributional RL, attributable to desirable smoothness properties of the distribution loss. We further characterize the effect of acceleration on distributional RL and discuss when distributional RL algorithms are effective in various environments. After gaining insights from the theoretical advantages of distributional RL, we further propose a novel distributional RL algorithm based on *Sinkhorn loss* that interpolates between the Wasserstein distance and maximum mean discrepancy (MMD). Our approach allows us to find a trade-off that simultaneously leverages the geometry of the Wasserstein distance and the favorable high-dimensional sample complexity and unbiased gradient estimates of MMD. In summary, our analysis opens the door to a deeper understanding of theoretical advantage of distributional RL.

## 2 PRELIMINARY KNOWLEDGE

In the standard RL setting, an agent interacts with an environment via a Markov decision process (MDP), a 5-tuple $(\mathcal{S}, \mathcal{A}, R, P, \gamma)$, where $\mathcal{S}$ and $\mathcal{A}$ are the state and action spaces, respectively. $P$ is the environment transition dynamics, $R$ is the reward function and $\gamma \in (0, 1)$ is the discount factor.

**State-value function vs. state-value distribution.** Given a policy $\pi$, the discounted sum of future rewards is a random variable $Z^\pi(s, a) = \sum_{t=0}^{\infty} \gamma^t R(s_t, a_t)$, where $s_0 = s$, $a_0 = a$, $s_{t+1} \sim P(\cdot|s_t, a_t)$, and $a_t \sim \pi(\cdot|s_t)$. In the control setting, expectation-based RL focuses on the state-value function $Q^\pi(s, a)$, which is the expectation of $Z^\pi(s, a)$, i.e., $Q^\pi(s, a) = \mathbb{E}[Z^\pi(s, a)]$. Distributional RL, on the other hand, focuses on the state-value distribution, the full distribution of $Z^\pi(s, a)$. Leveraging knowledge on the entire distribution can better capture the uncertainty of returns in the MDP beyond the expectation of return (Dabney et al., 2018a; Mavrin et al., 2019).

**Bellman operators vs. distributional Bellman operators.** For the policy evaluation in expectation-based RL, the value function is updated via the Bellman operator $\mathcal{T}^\pi Q(s, a) = \mathbb{E}[R(s, a)] + \gamma \mathbb{E}_{s' \sim p, \pi}[Q(s', a')]$. In distributional RL, the state-value distribution of $Z^\pi(s, a)$ is updated via the distributional Bellman operator $\mathfrak{T}^\pi$

$$\mathfrak{T}^\pi Z(s, a) = R(s, a) + \gamma Z(s', a'), \tag{1}$$

where $s' \sim P(\cdot|s, a)$ and $a' \sim \pi(\cdot|s')$.

From a theoretical perspective, both the Bellman operator $\mathcal{T}^\pi$ in the policy evaluation setting and the Bellman optimality operator $\mathcal{T}$ in the control setting are contractive in the stationary policy case. In contrast, the distributional Bellman operator $\mathfrak{T}^\pi$ is contractive under certain distribution divergence metrics, but the distributional Bellman optimality operator $\mathfrak{T}$ can only converge to a set of optimal non-stationary value distributions in a weak sense (Elie & Arthur, 2020).

## 3 EFFECT OF REGULARIZATION ON DISTRIBUTIONAL RL

Although the theoretical framework of distributional RL in the tabular setting has been basically established mentioned in Section 2, the theoretical understanding of its advantages over expectation-based RL has been less studied. In this section, we attribute the superiority of distributional RL into its regularization effect.

### 3.1 DISTRIBUTIONAL RL: NEURAL Z-FITTED ITERATION

**Neural Q-Fitted Iteration.** In the function approximation setting, Deep Q Learning (Mnih et al., 2015) can be simplified into *Neural Q-Fitted Iteration* (Fan et al., 2020) under tricks of experience replay and the target network $Q_{\theta*}$, where we update parameterized $Q_\theta(s, a)$ in each iteration $k$:

$$Q_\theta^{k+1} = \underset{Q_\theta}{\operatorname{argmin}} \frac{1}{n} \sum_{i=1}^{n} \left[ y_i - Q_\theta^k (s_i, a_i) \right]^2, \tag{2}$$

where the target $y_i = r(s_i, a_i) + \gamma \max_{a \in \mathcal{A}} Q_{\theta*}^k (s_i', a)$ is fixed within every $T_{\text{target}}$ steps to update target network $Q_{\theta*}$ by letting $\theta* = \theta$ and the experience buffer induces independent samples $\{(s_i, a_i, r_i, s_i')\}_{i \in [n]}$. In an ideal case that neglects the non-convexity and TD approximation errors, we have $Q_\theta^{k+1} = \mathcal{T} Q_\theta^k$, which is exactly the updating under Bellman optimality operator. Under the two conditions, the optimization problem in Eq. 2 can be viewed as Least Square Estimation (LSE) in a neural network parametric regression problem between the updating of target network $Q_{\theta*}$.

**Neural Z-Fitted Iteration.** Analogous to neural Q-fitted iteration, we can also simplify value-based distributional RL methods based on a parameterized $Z_\theta$ into a *Neural Z-fitted Iteration* as

$$Z_\theta^{k+1} = \underset{Z_\theta}{\operatorname{argmin}} \frac{1}{n} \sum_{i=1}^{n} d_p(Y_i, Z_\theta^k (s_i, a_i)), \tag{3}$$

where the target $Y_i = R(s_i, a_i) + \gamma Z_{\theta*}^k (s_i', \pi_Z(s')) $ with $\pi_Z(s') = \operatorname{argmax}_{a'} \mathbb{E}\left[ Z_{\theta*}^k(s', a') \right]$ is fixed within every $T_{\text{target}}$ steps to update target network $Z_{\theta*}$, and $d_p$ is a divergence metric between two distributions. Notably, the options of representation manner on $Z_\theta$ and the metric $d_p$ are pivotal for the empirical success of distributional RL algorithms. For instance, QR-DQN (Dabney et al., 2018b) approximates Wasserstein distance $W_p$, which leverages quantiles to represent the distribution of $Z_\theta$. C51 (Bellemare et al., 2017a) represents $Z_\theta$ via a categorical distribution under the convergence of Cramér distance (Bellemare et al., 2017b; Rowland et al., 2018), a special case with $p = 2$ of the $\ell_p$ distance (Elie & Arthur, 2020), while Moment Matching (Nguyen et al., 2020) learns deterministic samples to represent the distribution of $Z_\theta$ based on Maximum Mean Discrepancy (MMD). Contractive properties under typical metrics $d_p$ can be summarized as follows

- $\mathcal{T}^\pi$ is $\gamma$-contractive under the supreme form of Wassertein distance $W_p$.
- $\mathcal{T}^\pi$ is $\gamma^{1/p}$-contractive under the supreme form of $\ell_p$ distance.
- $\mathcal{T}^\pi$ is $\gamma^{\alpha/2}$-contractive under $\text{MMD}_\infty$ with the kernel $k_\alpha(x, y) = -\|x - y\|^\alpha, \forall \alpha \in \mathbb{R}$.

For the completeness, the definition of mentioned distances and the proof of contraction are provided in Appendix A. Although the widely used Kullback–Leibler (KL) divergence is not a contraction (Morimura et al., 2012), we show in Proposition 1 that the KL divergence still enjoys desirable properties in distributional RL context, which can be reasonable for the theoretical analysis. We assume $Z_\theta$ is absolutely continuous and has joint supports, under which the KL divergence is well-defined. Proof of Proposition 1 and the definition of supreme $D_{\text{KL}}$ are provided in Appendix B.

**Proposition 1.** *Denote the supreme of $D_{KL}$ as $D_{KL}^\infty$, we have: (1) $\mathfrak{T}^\pi$ is a non-expansive operator under $D_{KL}^\infty$, i.e., $D_{KL}^\infty(\mathfrak{T}^\pi Z_1, \mathfrak{T}^\pi Z_2) \leq D_{KL}^\infty(Z_1, Z_2)$, (2) $D_{KL}^\infty(Z_n, Z) \to 0$ implies $W_p(Z_n, Z) \to 0$, (3) the expectation of $Z^\pi$ is still $\gamma$-contractive, i.e., $\|\mathbb{E}\mathfrak{T}^\pi Z_1 - \mathbb{E}\mathfrak{T}^\pi Z_2\|_\infty \leq \gamma \|\mathbb{E} Z_1 - \mathbb{E} Z_2\|_\infty$.*

### 3.2 DISTRIBUTIONAL RL: A NOVEL ENTROPY-REGULARIZED MLE

The reasonable properties of KL divergence in Proposition 1 allows us to leverage it to conduct the theoretical analysis. To separate the impact of additional distribution information from the expectation of $Z^\pi$, we leverage the variant technique of *gross error model* from robust statistics (Huber, 2004), similar to the technique to analyze Label Smoothing (Müller et al., 2019) and Knowledge Distillation (Hinton et al., 2015). Specifically, we denote the one-dimensional full distribution of $Z^\pi$ as $F$, and the distribution on the remaining support getting rid of $\mathbb{E}[Z^\pi]$ as $F_\mu$. Hence, we can obtain the distribution decomposition for $Z^\pi(s, a)$ as

$$F^{s,a}(x) = (1 - \epsilon) \mathbb{1}_{\{x \geq \mathbb{E}[Z^\pi(s,a)]\}}(x) + \epsilon F_\mu^{s,a}(x), \tag{4}$$

where $\epsilon$ controls the proportion of $F_\mu^{s,a}(x)$ and the indicator function $\mathbb{1}_{\{x \geq \mathbb{E}[Z^\pi(s,a)]\}} = 1$ if $x \geq \mathbb{E}[Z^\pi(s,a)]$, otherwise 0. After taking derivatives on both sides, we attain the relationship of their density functions as $p^{s,a}(x) = (1 - \epsilon)\delta_{\{x = \mathbb{E}[Z^\pi(s,a)]\}}(x) + \epsilon\mu^{s,a}(x)$, where $\mu^{s,a}(x)$ is the density function related to $Z^\pi(s,a)$ on remaining supports removing $\mathbb{E}[Z^\pi(s,a)]$. It is worth noting that the existence of $\mu^{s,a}(x)$ can be simply guaranteed by directly computing $\mu^{s,a}(x) = p^{s,a}(x)/\epsilon - (1 - \epsilon)\delta_{\{x = \mathbb{E}[Z^\pi(s,a)]\}}/\epsilon$ as long as $p^{s,a}(x)$ and the expectation of $Z^\pi(s,a)$ exist. Next, we use $p^{s,a}(x)$ and $q_\theta^{s,a}(x)$ to denote the density distributions behind $\{Y_i\}_{i \in [n]}$ and $Z_\theta^k(s,a)$ in neural Z-fitted iteration via Eq. 3, respectively. Therefore, we can derive the following result in Proposition 2.

**Proposition 2.** *Let $\mathcal{H}(P, Q)$ as the cross entropy, i.e., $\mathcal{H}(P, Q) = -\int_{x \in \mathcal{X}} P(x) \log Q(x)\, \mathrm{d}x$. Let $\alpha$ be a positive constant, and based on the decomposition in Eq. 4 and $D_{KL}$ as $d_p$, Neural Z-fitted iteration in Eq. 3 can be reformulated as*

$$Z_\theta^{k+1} = \underset{Z_\theta}{\operatorname{argmin}} \frac{1}{n} \sum_{i=1}^n \mathcal{H}(\delta_{\{x = \mathbb{E}[Z^\pi(s_i,a_i)]\}}, q_\theta^{s_i,a_i}) + \alpha \mathcal{H}(\mu^{s_i,a_i}, q_\theta^{s_i,a_i}). \tag{5}$$

We provide the proof in Appendix C. For the uniformity of notation, we still use $s, a$ in the following analysis instead of $s_i, a_i$ in Eq. 5. Importantly, the first term in Eq. 5 can be further simplified as $-\int_{x \in \mathcal{X}} \log q_\theta^{s,a}(\mathbb{E}[Z(s,a)])$. Minimizing this first term can be viewed as a variant of Maximum Likelihood Estimation (MLE) on the *expectation* $\mathbb{E}[Z(s,a)]$ rather than the traditional MLE directly on *observed samples*. The cross entropy regularization in the second term pushes $q_\theta^{s,a}$ to approximate the distribution $\mu^{s,a}$ in order to fully utilize the additional distributional information while learning, serving as the key to the superiority of distributional RL. This novel cross entropy regularization regarding $\mu^{s,a}$ and $q_\theta^{s,a}$ is different from the classical entropy regularization used in RL, which we further analyze their connection and discrepancy in Section 3.3. In summary, distributional RL can be simplified as a novel entropy-regularized MLE within neural Z-fitted iteration framework in stark contrast to the Least-Square estimation of expectation-based RL in the neural Q-fitted iteration.

### 3.3 CONNECTION WITH MAXIMUM ENTROPY RL

We establish the connection between the derived novel cross entropy regularization in Eq. 5 in distributional RL with the classical maximum entropy RL (Williams & Peng, 1991). Maximum entropy RL, including Soft Q-Learning (Haarnoja et al., 2017), greedily maximizes the entropy of the policy $\pi(\cdot|s)$ in each state:

$$J(\pi) = \sum_{t=0}^T \mathbb{E}_{(s_t,a_t) \sim \rho_\pi} \left[ r(s_t, a_t) + \beta \mathcal{H}(\pi(\cdot|s_t)) \right], \tag{6}$$

where $\mathcal{H}(\pi_\theta(\cdot|s_t)) = -\sum_a \pi_\theta(a|s_t) \log \pi_\theta(a|s_t)$ and $\rho_\pi$ is the generated distribution following $\pi$. The temperature parameter $\beta$ determines the relative importance of the entropy term against the cumulative rewards, and thus controls the stochasticity of the optimal policy. This maximum entropy regularization has various conceptual and practical advantages. Firstly, the policy encourages exploration, avoiding situations in which the agent might fall into a local optimum. Secondly, it considerably improves learning speed over classical methods and therefore are widely used in state-of-the-art algorithms, e.g., Soft Actor-Critic (SAC) (Haarnoja et al., 2018) and PPO (Schulman et al., 2017). Lastly, maximum entropy can yield more robustness to abnormal or rare events while developing a task. Similar robustness superiority of distributional RL against noisy state observations has been investigated in (Sun et al., 2021).

Similar benefits of both distributional RL and maximum entropy RL motivate us to explore their mathematical connection, and we refocus on the Bellman updating based on these two kinds of entropy-based regularization. In particular, *Soft Policy Evaluation* proposed in SAC has shown that by alternately iterating the following equations, the algorithm is equivalent to maximize the maximum entropy RL objective function in Eq. 6: $\mathcal{T}^\pi Q(s_t, a_t) \triangleq r(s_t, a_t) + \gamma \mathbb{E}_{s_{t+1} \sim \rho^\pi}[V(s_{t+1})]$, where $V(s_{t+1}) = \mathbb{E}_{a_{t+1} \sim \pi}[Q(s_{t+1}, a_{t+1}) - \log \pi(a_{t+1}|s_{t+1})]$ is the soft value function (Haarnoja et al., 2018). Interestingly, we still have the similar convergence result related to our novel cross entropy regularization in Eq. 5, which is shown in Theorem 1:

**Theorem 1.** *Consider the soft Bellman operator $\mathcal{T}_d^\pi$ in Eq. 7 and a mapping $Q^0 : \mathcal{S} \times \mathcal{A} \to \mathbb{R}$ with $|\mathcal{A}| \leq \infty$, and define $Q^{k+1} = \mathcal{T}_d^\pi Q^k$. Given the true distribution $\mu^{s_t,a_t}$ for each t, then the*

*sequence $Q^k$ will converge to a soft Q-value of $\pi$ as $k \to \infty$ with the new entropy objective function as $J'(\theta) = \sum_{t=0}^{T} \mathbb{E}_{(s_t, a_t) \sim \rho^\pi} \left[ r(s_t, a_t) - \gamma \mathcal{H}(\mu^{s_t, a_t}, q_\theta^{s_t, a_t}) \right]$:*

$$\mathcal{T}_d^\pi Q(s_t, a_t) \triangleq r(s_t, a_t) + \gamma \mathbb{E}_{s_{t+1} \sim \rho^\pi} \left[ V(s_{t+1}) \right], \tag{7}$$

*where $V(s_{t+1}) = \mathbb{E}_{a_{t+1} \sim \pi, x \sim \mu^{s_{t+1}, a_{t+1}}} \left[ Q(s_{t+1}, a_{t+1}) + \log q_\theta^{s_{t+1}, a_{t+1}}(x) \right]$ and the policy $\pi$ follows the classical greedy policy rule, i.e., $\pi(\cdot|s) = \arg\max_a \mathbb{E}[Z_\theta(s, a)]$, determined by $\theta$.*

Please refer to Appendix D for the proof. The new entropy augmented objective function $J'(\theta)$ in Theorem 1 reveals that different from adding the entropy regarding the policy $\pi(\cdot|s_t)$ in maximum entropy RL, distributional RL can be viewed as subtracting a cross entropy term between $\mu$ and $q_\theta$. Moreover, the entropy in maximum entropy RL is state-wise, while our cross entropy regularization is state-action-wise, which is a more fine-grained characterization on the variability of the full distribution of return $Z$.

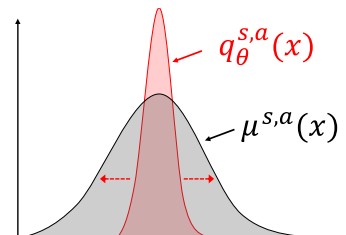

Notably, we highlight that our derived cross entropy regularization is *risk-aware*. If the true state-action distribution $F_\mu$ has higher degree of dispersion, e.g., a larger variance, than the current $q_\theta$, minimizing our entropy automatically encourages the policy to explore the uncertainty of the environment through changing $Z_\theta$. A sketch map in Figure 1 is provided to illustrate the impact of our risk-aware entropy regularization. We remark that this

Figure 1: Impact of the risk-aware regularization in distributional RL. The entropy-based regularization will push $q_\theta^{s,a}$ to match the true density function $\mu^{s,a}$ determined by the environment, and thus let the learned policy be fully aware of the uncertainty of environment.

risk-aware regularization may not necessarily encourage exploration under the greedy action rule $\pi_Z(s) = \arg\max_a \mathbb{E}[Z_\theta(s, a)]$ even though $q_\theta$ changes towards $\mu$ as exhibited in Figure 1. A risk-aware action rule on the current greedy version might be preferred in the future.

## 4  STABLE OPTIMIZATION AND GUARANTEED GENERALIZATION

In this section, we further characterize $q_\theta^{s,a}$ in order to attain stable optimization and guaranteed generalization properties of distributional RL. Concretely, we leverage the histogram function $f^{s,a}$ to parameterize the density function of $Z_\theta(s, a)$, i.e., $q_\theta^{s,a}$, still based on the KL divergence as $d_p$, yielding the *histogram distributional loss* (Imani & White, 2018) within each update in the neural Z-fitted iteration. Denote $\mathbf{x}(s)$ as the state feature on each state $s$, and we let the support of $\mathbf{x}(s)$ be uniformly partitioned into $k$ bins. We let the function $f^{s,a} : \mathcal{X} \to [0, 1]^k$ provide a k-dimensional vector $f^{s,a}(\mathbf{x}(s))$ of the coefficients indicating the probability the target is in that bin given the state $s$ and action $a$ pair, and $\mathbf{x}(s)$. We use *softmax* based on the linear approximation $\mathbf{x}(s)^\top \theta_i$ to express $f^{s,a}$, i.e., $f_i^{s,a,\theta}(\mathbf{x}(s)) = \exp(\mathbf{x}(s)^\top \theta_i) / \sum_{j=1}^{k} \exp(\mathbf{x}(s)^\top \theta_j)$. For simplicity, we use $f_i^\theta(\mathbf{x}(s))$ to replace $f_i^{s,a,\theta}(\mathbf{x}(s))$. Therefore, the resulting *histogram distributional loss* $\mathcal{L}_\theta$ is formulated as:

$$\mathcal{L}_\theta(s, a) = D_{\mathrm{KL}}(p^{s,a}, q_\theta^{s,a}) = \int_{x \in \mathcal{X}} p^{s,a}(x) \log(p^{s,a}(x)/q_\theta^{s,a}(x)) \, dx = -\sum_{i=1}^{k} p_i^{s,a} \log f_i^\theta(\mathbf{x}(s))$$
$$\tag{8}$$

where $\theta = \{\theta_1, ..., \theta_k\}$ and $p_i^{s,a}$ is the histogram probability of $p^{s,a}(x)$ defined in Eq. 4. The derivation of the histogram distributional loss is given in Appendix E. To attain the stable optimization property of distributional RL, we firstly derive Lemma 1 as follows

**Lemma 1.** *(Properties of Histogram Distributional Loss) Assume each state feature is bounded, i.e., $\|\mathbf{x}(s)\| \le l$, then $\mathcal{L}_\theta$ is $kl^2$-smooth, convex and $kl$-Lipschitz continuous w.r.t. $\theta$.*

Please refer to Appendix F for the proof. The derived Lipschitz properties of $d_p$ under histogram distributional loss plays an integral part in the stable optimization of distributional RL, but the classical Least-Squared estimation in neural-Q-fitted iteration for expectation-based RL does not enjoy these smoothness properties. In particular, for histogram distributional loss we have $\|\nabla_\theta \mathcal{L}_\theta\| \le kl$, while the counterpart in expectation-based is $|y_i - Q_\theta^k(s, a)| \|\mathbf{x}(s)\|$, which might be arbitrarily large

due to the unbounded $|y_i - Q_\theta^k(s,a)|$. To further derive the uniform stability of distributional RL while running *stochastic gradient descend* (SGD), we introduce its definition in Definition 1.

**Definition 1.** *(Uniform Stability) (Hardt et al., 2016) Consider a function $g$, a randomized algorithm $\mathcal{M}$ is uniformly stable if for all data sets $S, S'$ such that $S, S'$ differ in at most one example, we have*

$$\sup_x \mathbb{E}_\mathcal{M} \left[ g(\mathcal{M}(S); x) - g(\mathcal{M}(S'); x) \right] \leq \epsilon_{stab}. \tag{9}$$

Given the definition of uniform stability, we show that distributional RL under histogram distributional loss while running SGD is $\epsilon_{\text{stab}}$-uniformly stable and generalization guaranteed in Theorem 2.

**Theorem 2.** *(Stable Optimization and Guaranteed Generalization) Suppose that we run SGD under $\mathcal{L}_\theta$ in Eq. 8 with step sizes $\lambda_t \leq 2/kl^2$ for $T$ steps, and $\|\mathbf{x}(s)\| \leq l$ for each state $s$, we have:*

*(1) SGD under $\mathcal{L}_\theta$ satisfies the uniform stability in Definition 1 with $\epsilon_{stab} \leq \frac{4kT}{n}$,*

*(2) Suppose that the data distribution is stable within each neural Z-fitted iteration, the generalization gap for solving the entropy regularized MLE in Eq. 5 is $\epsilon_{stab}$-bounded.*

Please refer to the proof of Theorem 2 in Appendix G. In summary, the optimization to solve the entropy regularized MLE of distributional RL within the update of neural Z-fitted iteration is stable with the stability errors shrinking at the rate of $O(n^{-1})$. This stability optimization properties is ascribed to the smooth and convex properties from histogram distributional loss (non-convex case still holds in (Hardt et al., 2016)) and can further control the generalization gap. By contrast, the least-estimation in neural Q-fitted iteration for expectation-based RL, which is without these smooth properties, can not yield the stable optimization and guaranteed generalization directly.

## 5 ACCELERATION EFFECT

Based on the setting in Section 4, we further incorporate the decomposition of $F^{s,a}(x)$ proposed in Eq. 4 into the analysis, where $p^{s,a}(x) = (1 - \epsilon)\delta_{\{x=\mathbb{E}[Z^\pi(s,a)]\}}(x) + \epsilon\mu^{s,a}(x)$, in order to derive the acceleration effect of distributional RL. Within each update in the neural Z-fitted iteration, the target is to minimize $\frac{1}{n} \sum_{i=1}^n \mathcal{L}_\theta(s_i, a_i)$. We denote $G^k(\theta)$ as the expectation of $\mathcal{L}_\theta$, i.e., $G^k(\theta) = \mathbb{E}_{(s,a)\sim\rho^\pi} [\mathcal{L}_\theta(s,a)]$. As such, the convex and smooth properties in Lemma 1 still hold for $G^k(\theta)$. We use $G(\theta)$ for $G^k(\theta)$ for simplicity. As the KL divergence enjoys the property of unbiased gradient estimates, we let the variance of its stochastic gradient over *the expectation* be bounded, i.e., $\mathbb{E}_{(s,a)\sim\rho^\pi} \left[ \|\nabla\mathcal{L}_\theta(\delta_{\{x=\mathbb{E}[Z^\pi(s,a)]\}}, q_\theta^{s,a})) - \nabla G(\theta)\|^2 \right] = \sigma^2$. Next, we characterize the approximation degree of $q_\theta^{s,a}$ to the ground-truth distribution $\mu^{s,a}$ defined in Eq. 5 by measuring its variance as $\kappa\sigma^2$:

$$\mathbb{E}_{(s,a)\sim\rho^\pi} \left[ \|\nabla\mathcal{L}_\theta(\mu^{s,a}, q_\theta^{s,a})) - \nabla G(\theta)\|^2 \right] = \kappa\sigma^2. \tag{10}$$

Notably, a favorable approximation of $q_\theta^{s,a}$ to $\mu_\theta^{s,a}$ would lead to a small $\kappa$, in which case an acceleration effect of distributional RL can be derive as shown in Theorem 3. Based on Eq. 10, we immediately have the following lemma.

**Lemma 2.** *As $p^{s,a}(x) = (1 - \epsilon)\delta_{\{x=\mathbb{E}[Z^\pi(s,a)]\}}(x) + \epsilon\mu^{s,a}(x)$, we have:*

$$\mathbb{E}_{(s,a)\sim\rho^\pi} \left[ \|\nabla\mathcal{L}_\theta(p^{s,a}, q_\theta^{s,a})) - \nabla G(\theta)\|^2 \right] \leq (1 - \epsilon)\sigma^2 + \epsilon\kappa\sigma^2. \tag{11}$$

Please refer to Appendix H for the proof. To measure the convergence of the entropy regularized MLE in distributional RL or least-square estimation in expectation-based RL, we need the following definition of the first-order $\tau$-stationary point.

**Definition 2.** *(First-order $\tau$-Stationary Point) While solving $\min_\theta G(\theta)$, the updated parameters $\theta_T$ after $T$ steps is a first-order $\tau$-stationary point if $\|\nabla G(\theta_T)\| \leq \tau$, where the small $\tau$ is in $(0, 1)$.*

We formally show our Theorem 3 to characterize the acceleration effect of distributional RL determined by the variance magnitude $\tau$.

**Theorem 3.** *(Acceleration Effect) While running SGD to solve the entropy-regularized MLE within neural Z-fitted iteration in Eq. 5 with the step size $\lambda = 1/kl^2$ and $\epsilon = 1/(1 + \tau)$, we have:*

*(1) The sample complexity is $O(\frac{1}{\tau^4})$ if we only consider the expectation in $p^{s,a}$, i.e., $\delta_{\{x=\mathbb{E}[Z^\pi(s,a)]\}}$.*

*(2) When $\kappa \leq \frac{\tau^2}{4\sigma^2}$ and let $T = \frac{4G(\theta_0)}{\lambda \tau^2}$, the regularized MLE of distributional RL within each neural Z-fitted iteration converges to a $\tau$-stationary point in expectation with sample complexity $O(\frac{1}{\tau^2})$.*

*(3) When $\kappa > \frac{\tau^2}{4\sigma^2}$ and let $T = \frac{G(\theta_0)}{\lambda \kappa \tau^2}$, the regularized MLE of distributional RL does not converge to a $\tau$-stationary point, but we have $\mathbb{E}\left[\|\nabla G(\theta)\|^2\right] \leq O(\kappa)$.*

The proof is provided in Appendix I. Theorem 3 is inspired by (Xu et al., 2020) to analyze the convergence of label smoothing, which can be similarly utilized to analyze the acceleration effect of distributional RL. Importantly, we need to emphasize that Theorem 3 in fact reveals the reason why distributional RL algorithms can achieve inconsistent superiority across different Atari games. This phenomenon is mainly owing to the different degree of TD target approximation to $p^{s,a}$, especially $\mu^{s,a}$ for various environments. In particular, a small approximation error of $q_\theta$ to $p$ (or $\mu$) corresponds to a small $\kappa$, This can be normally $\leq \tau^2/4\sigma^2$, yielding better sample efficiency and accelerating the algorithm relative to expectation-based RL. Conversely, an unsatisfactory approximation may not lead to the acceleration effect, but the bounded gradient normally corresponds to a reasonable performance, coinciding with previous empirical observations on various environments.

**Extension of Representation Effect.** Apart from the acceleration effect, we also conduct some empirical analysis of distributional RL from the perspective of representation. In particular, distributional RL encourages state representation from the same action class classified by the policy in tighter clusters. Please refer to Appendix K for more details.

## 6 ALGORITHM: SINKHORN DISTRIBUTIONAL RL

Through the theoretical analysis, we can gain insights into the advantages of distribution RL over expectation-based RL, attributing to entropy-based regularization, stable optimization and acceleration effect determined by the distributional approximation error (measured by $\kappa$). From the algorithmic perspective, the choice of representation manner on $Z_\theta$ and the distributional divergence metric $d_p$ mentioned in Section 3.1 are pivotal for the final performance of distributional RL algorithms.

we further design Sinkhorn distributional RL algorithm. Sinkhorn loss (Sinkhorn, 1967) is a tractable loss to approximate optimal transport problem by leveraging an entropic regularization to turn the original Wasserstein distance into a differentiable and more robust quantity. The resulting loss can be computed using Sinkhorn fixed point iterations, which is naturally suitable for modern deep learning frameworks. In particular, the entropic smoothing generates a family of losses interpolating between Wasserstein distance and Maximum Mean Discrepancy (MMD). Thus it allows us to find a sweet trade-off that simultaneously leverages the geometry of Wasserstein distance on the one hand, and the favorable high-dimensional sample complexity and unbiased gradient estimates of MMD. We introduce the entropic regularized Wassertein distance as

$$\min_{\Pi \in \mathbf{\Pi}(u,v)} \int c(x,y)\mathrm{d}\Pi(x,y) + \varepsilon \int \log\left(\frac{\Pi(x,y)}{\mathrm{d}u(x)\mathrm{d}v(y)}\right)\mathrm{d}\Pi(x,y), \tag{12}$$

where $c$ is the cost function in optimal transport. This objective function associated with cost function $c$ can be rewritten as $\mathcal{W}_{c,\varepsilon}(\mu,\nu) = \int c(x,y)\mathrm{d}\Pi_\varepsilon(x,y)$. Therefore, the sinkhorn loss between two measures $u$ and $v$ is defined as

$$\overline{\mathcal{W}}_{c,\varepsilon}(u,v) = 2\mathcal{W}_{c,\varepsilon}(u,v) - \mathcal{W}_{c,\varepsilon}(u,u) - \mathcal{W}_{c,\varepsilon}(v,v) \tag{13}$$

**Theorem 4.** *If we leverage Sinkhorn loss as the metric in distributional RL and choose $c$ as unrectified kernel $k_\alpha$, i.e., $k_\alpha(x,y) := -\|x-y\|^\alpha$, for $\forall \alpha \in \mathbb{R}$, we have:*

*(1) As $\epsilon \to 0$, $\overline{\mathcal{W}}_{c,\varepsilon}(u,v) \to 2\mathcal{W}_\alpha(u,v)$, and thus $\mathfrak{T}^\pi$ is a $\gamma$-contraction.*

*(2) As $\epsilon \to \infty$, $\overline{\mathcal{W}}_{c,\varepsilon}(u,v) \to MMD_{-k_\alpha}(u,v)$, and thus $\mathfrak{T}^\pi$ is a $\gamma^{\alpha/2}$-contraction.*

Proof is provided in Appendix J. Theorem 4 indicates that if we choose $c$ as the *unrectified* kernel, the limiting behaviors of distributional Bellman operator $\mathfrak{T}^\pi$ are both contractive under Sinkhorn loss. Note that without the limitation, the entropic regularization restricts the search space on the optimal

transport that eventually corresponds to a Gibbs kernel, based on which $\mathfrak{T}^\pi$ may not be a contraction. However, similar to MMD method, we show that our approach can still achieve empirical success and is very competitive across a wide range of Atari games in Section 7.

In the algorithm design, similar to in MMD distributional RL (Nguyen et al., 2020), we apply particle representation to represent $Z_\theta(s, a)$ by directly generating multiple deterministic samples. Finally, we minimize the Sinkhorn loss between the approximate distribution via multiple samples and its distributional Bellman target. A detailed description of algorithm is provided in Algorithm 1.

**Remark.** MMD enjoys a convergence rate of $O(n^{-1/2})$ 1 regardless of the underlying dimension while 1-Wasserstein distance has a convergence rate of $O(n^{-1/d})$ if $d > 2$, which is slower for large $d$. Thus, Sinkhorn loss has the potential to enjoy the faster convergence of MMD.

---

**Algorithm 1** Generic Sinkhorn Algorithm

**Require**: Number of particles $N$, number of Sinkhorn iteration $L$ and hyperparameter $\varepsilon$.
**Input**: Sample transition $(s, a, r', s')$
  1: **if** Policy evaluation **then**
  2:     $a^* \sim \pi(\cdot|s')$.
  3: **else**
  4:     $a^* \leftarrow \arg\max_{a' \in \mathcal{A}} \frac{1}{N} \sum_{i=1}^{N} Z_\theta\left(s', a'\right)_i$
  5: **end if**
  6: $\mathfrak{T}Z_i \leftarrow r + \gamma Z_{\theta^-}\left(s', a^*\right)_i, \forall 1 \le i \le N$
**Output**: $\overline{\mathcal{W}}_{k_2, \varepsilon}\left(\{Z_\theta(s, a)_i\}_{i=1}^N, \{\mathfrak{T}Z_i\}_{i=1}^N\right)$

---

## 7 EXPERIMENTS

We demonstrate the effectiveness of Sinkhorn distributinal RL (SinkhornDRL) as described in Algorithm 1 on a wide range of Atari 2600 games. Specifically, we leverage the same architecture as QR-DQN (Dabney et al., 2018b) for a fair comparison. More advanced techniques that can expand the model expressiveness from IQN (Dabney et al., 2018a) and FQF (Yang et al., 2019) can be naturally incorporated into our framework.

**Baselines.** Due to interpolation characteristic of SinkhornDRL, we choose 3 typical distributional RL algorithms as classic baselines, including QR-DQN (Dabney et al., 2018b), C51 (Bellemare et al., 2017a) and MMD (Nguyen et al., 2020), as well as DQN (Mnih et al., 2015). MMD algorithm is implemented with the same architecture as QRDQN, and leverages Gaussian kernels $k_h(x, y) = $

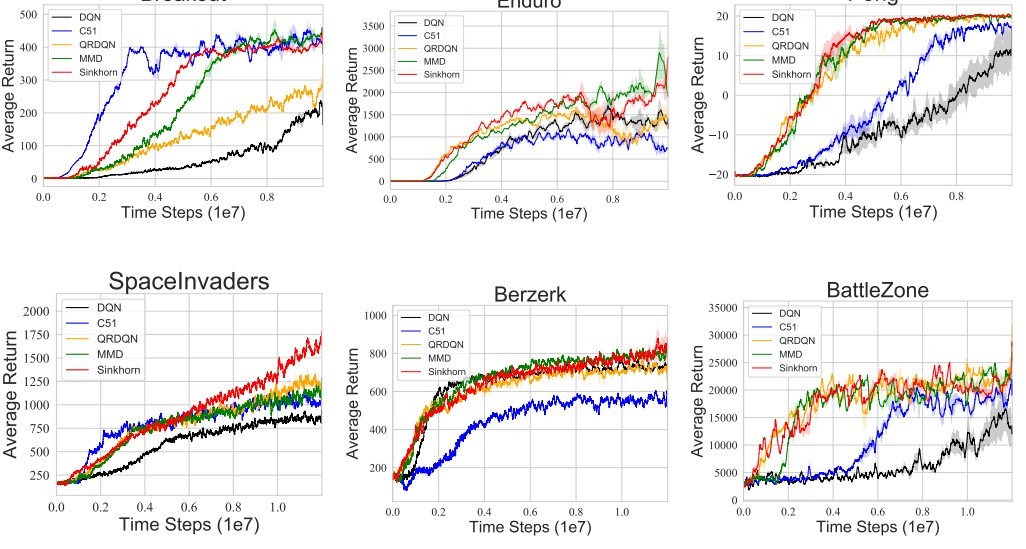

Figure 2: Performance of SinkhornDRL on six Atari games.

$\exp(-(x-y)^2/h)$ with the kernel mixture trick covering a range of bandwidths $h$, which is same as the basic setting in the original MMD paper (Nguyen et al., 2020).

**Hyperparameter settings.** For a fair comparison with QR-DQN, C51 and MMD, we used the same hyperparamters: the number of generated samples $N = 200$, Adam optimizer with lr $= 0.00005, \epsilon_{\text{Adam}} = 0.01/32$. We used a target network to compute the distributional Bellman target, which fits well in the neural Z-fitted iteration framework. In addition, we choose number of Sinkhorn iterations $L = 10$ and smoothing hyperparameter $\varepsilon = 10.0$ in Section 7.1 as they are not sensitive within a proper interval demonstrated in Section 7.2. We choose $\alpha = 2$ in $k_\alpha$.

## 7.1 PERFORMANCE OF SINKHORNDRL

Figure 2 illustrates that SinkhornDR can achieve the state-of-the-art or competitive performance on typical Atari games compared with various baseline algorithms (DQN, QR-DQN, C51, MMD) in different metrics $d_p$ and representation manners on $Z_\theta$. Even though C51 outperforms others on breakout game, it is significantly inferior on other games, e.g., Enduro, Pong and Berzerk. Moreover, SinkhornDRL significantly outperforms MMD on breakout and SpaceInvader, and this superiority can be owing to the theoretical advantage of Sinkhorn loss over MMD. This coincides with the theoretical results as demonstrated in Theorem 4 that Sinkhorn loss interpolates between Wasserstein distance and MMD, which can simultaneously make full use of the data geometry from Wasserstein distance and the faster convergence, unbiased gradient estimates from Maximum Mean Discrepancy. We provide the competitive performance of SinkhornDRL on other Atari games in Appendix L.

## 7.2 SENSITIVITY ANALYSIS

We further examine the impact of different hyperparameters in SinkhornDRL on the final performance, including the smoothing hyperparameter $\varepsilon$ in Eq. 12 the number of deterministic samples $N$ in Algorithm 1. From the left diagram in Figure 3, we can observe that our algorithm is not sensitive to the magnitude of $\varepsilon$ as long as $\varepsilon$ is within an appropriate interval, e.g., $[1, 100]$. Mean-

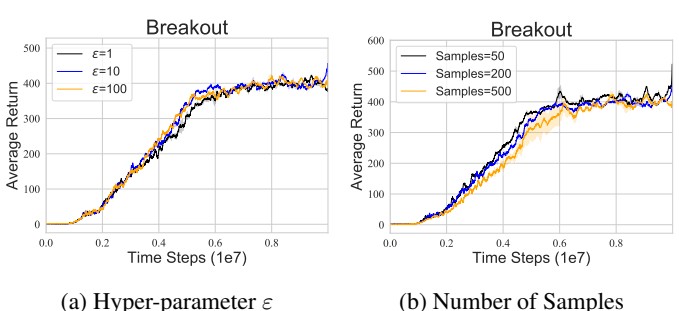

(a) Hyper-parameter $\varepsilon$      (b) Number of Samples

Figure 3: Sensitivity analysis of SinkhornDRL on Breakout.

while, it turns out that an overly large number of samples $N$ can even slightly worsen the performance of SinkhornDRL.

## 8 DISCUSSIONS AND CONCLUSION

Implicit generative models can be further incorporated into SinkhornDRL, including parameterizing the cost function in Sinkhorn loss, which can be naturally expected to achieve more promising performance in the future. Moreover, a direct analysis on Wasserstein or $\ell_p$ distance can be more effective than KL divergence presented in this paper, albeit being theoretically tricky.

In this paper, we illuminate the superiority of distributional RL over expectation-based RL from the perspectives of regularization, optimization, acceleration and representation. In addition, a novel family of distributional RL algorithms based on Sinkhorn loss is designed that accomplishes the promising performance on Atari games. Our analysis paves the way towards deeper understanding of distributional RL, and further promote its deployment in real applications.

**Ethics Statement.** Revealing the advantage of distributional RL would promote the application of distributional RL algorithms in real scenarios. As distributional RL enjoys the robustness against

abnormal events, e.g., noisy state observations, it can also be beneficial for the privacy of algorithms. Besides, the deeper insights into distributional RL plays a key role into the research integrity issue. Based on our knowledge, it is not related to harmful applications or fairness issues.

**Reproducibility Statement.** For the theoretical part, we clearly state the related assumption and detailed proof process in the appendix. In terms of the algorithm implementation, our Sinkhorn algorithm is directly adapted from the public distributional RL algorithms, such as MMD.

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

## A    DEFINITION OF DISTANCES AND CONTRACTION

**Definition of distances.** Given two random variables $X$ and $Y$, $p$-Wasserstein metric $W_p$ between the distributions of $X$ and $Y$ is defined as

$$W_p(X,Y) = \left( \int_0^1 \left| F_X^{-1}(\omega) - F_Y^{-1}(\omega) \right|^p d\omega \right)^{1/p} = \| F_X^{-1} - F_Y^{-1} \|_p, \tag{14}$$

which $F^{-1}$ is the inverse cumulative distribution function of a random variable with the cumulative distribution function as $F$. Further, $\ell_p$ distance (Elie & Arthur, 2020) is defined as

$$\ell_p(X,Y) := \left( \int_{-\infty}^{\infty} \left| F_X(\omega) - F_Y(\omega) \right|^p d\omega \right)^{1/p} = \| F_X - F_Y \|_p \tag{15}$$

The $\ell_p$ distance and Wasserstein metric are identical at $p = 1$, but are otherwise distinct. Note that when $p = 2$, $\ell_p$ distance is also called Cramér distance (Bellemare et al., 2017b) $d_C(X, Y)$. Also, the Cramér distance has a different representation given by

$$d_C(X,Y) = \mathbb{E}|X - Y| - \frac{1}{2}\mathbb{E}\left|X - X'\right| - \frac{1}{2}\mathbb{E}\left|Y - Y'\right|, \tag{16}$$

where $X'$ and $Y'$ are the i.i.d. copies of $X$ and $Y$. Energy distance (Székely, 2003; Ziel, 2020) is a natural extension of Cramér distance to the multivariate case, which is defined as

$$d_E(\mathbf{X}, \mathbf{Y}) = \mathbb{E}\|\mathbf{X} - \mathbf{Y}\|_2 - \frac{1}{2}\mathbb{E}\|\mathbf{X} - \mathbf{X}'\| - \frac{1}{2}\mathbb{E}\|\mathbf{Y} - \mathbf{Y}'\|, \tag{17}$$

where $\mathbf{X}$ and $\mathbf{Y}$ are multivariate. Moreover, the energy distance is a special case of the maximum mean discrepancy (MMD), which is formulated as

$$\mathrm{MMD}(\mathbf{X}, \mathbf{Y}; k) = \left( \mathbb{E}\left[ k\left(\mathbf{X}, \mathbf{X}'\right) \right] + \mathbb{E}\left[ k\left(\mathbf{Y}, \mathbf{Y}'\right) \right] - 2\mathbb{E}[k(\mathbf{X}, \mathbf{Y})] \right)^{1/2} \tag{18}$$

where $k(\cdot, \cdot)$ is a continuous kernel on $\mathcal{X}$. In particular, if $k$ is a trivial kernel, MMD degenerates to energy distance. Additionally, we further define the supreme MMD, which is a functional $\mathcal{P}(\mathcal{X})^{\mathcal{S} \times \mathcal{A}} \times \mathcal{P}(\mathcal{X})^{\mathcal{S} \times \mathcal{A}} \to \mathbb{R}$ defined as

$$\mathrm{MMD}_\infty(\mu, \nu) = \sup_{(x,a) \in \mathcal{S} \times \mathcal{A}} \mathrm{MMD}_\infty(\mu(x,a), \nu(x,a)) \tag{19}$$

**Proof of Contraction.**

- Contraction under supreme form of Wasserstein diatance is provided in Lemma 3 (Bellemare et al., 2017a).
- Contraction under supreme form of $\ell_p$ distance can refer to Theorem 3.4 (Elie & Arthur, 2020).
- Contraction under $\mathrm{MMD}_\infty$ is provided in Lemma 6 (Nguyen et al., 2020).

## B    PROOF OF PROPOSITION 1

*Proof.* (1) We recap three crucial properties of a divergence metric. The first is *scale sensitive* (**S**) (of order $\beta$, $beta > 0$), i.e., $d_p(cX, cY) \le |c|^\beta d_p(X, Y)$. The second property is *shift invariant* (**I**), i.e., $d_p(A + X, A + Y) \le d_p(X, Y)$. The last one is *unbiased gradient* (**U**). We use $p$ and $q$ to denote the density function of two random variables $X$ and $Y$, and thus $D_{\mathrm{KL}}(X, Y)$ is defined as $D_{\mathrm{KL}}(X, Y) = \int_{-\infty}^{\infty} p(x) \frac{p(x)}{q(x)} \, dx$. Firstly, we show that $D_{\mathrm{KL}}(X, Y)$ is NOT scale sensitive:

$$\begin{aligned} D_{\mathrm{KL}}(cX, cY) &= \int_{-\infty}^{\infty} \frac{1}{a} p(\frac{x}{a}) \frac{\frac{1}{a}p(\frac{x}{a})}{\frac{1}{a}q(\frac{x}{a})} \, dx \\ &= \int_{-\infty}^{\infty} p(y) \frac{p(y)}{q(y)} \, dy \\ &= D_{\mathrm{KL}}(X, Y), \text{ with } \beta = 0 \end{aligned} \tag{20}$$

We further show that $D_{\mathrm{KL}}(X, Y)$ is shift invariant:

$$
\begin{aligned}
D_{\mathrm{KL}}(A + X, A + Y) &= \int_{-\infty}^{\infty} p(x - A) \frac{p(x - A)}{q(x - A)} \, \mathrm{d}x \\
&= \int_{-\infty}^{\infty} p(y) \frac{p(y)}{q(y)} \, \mathrm{d}y \\
&= D_{\mathrm{KL}}(X, Y)
\end{aligned}
\tag{21}
$$

Moreover, it is well-known that KL divergence has unbiased sample gradients (Bellemare et al., 2017b). The supreme $D_{\mathrm{KL}}$ is a functional $\mathcal{P}(\mathcal{X})^{\mathcal{S} \times \mathcal{A}} \times \mathcal{P}(\mathcal{X})^{\mathcal{S} \times \mathcal{A}} \to \mathbb{R}$ defined as

$$
D_{\mathrm{KL}}^{\infty}(\mu, \nu) = \sup_{(x, a) \in \mathcal{S} \times \mathcal{A}} D_{\mathrm{KL}}(\mu(x, a), \nu(x, a))
\tag{22}
$$

Therefore, we prove $\mathfrak{T}^{\pi}$ is at best a non-expansive operator under the supreme form of $D_{\mathrm{KL}}$:

$$
\begin{aligned}
&D_{\mathrm{KL}}^{\infty}(\mathfrak{T}^{\pi} Z_1, \mathfrak{T}^{\pi} Z_2) \\
&= \sup_{s, a} D_{\mathrm{KL}}(\mathfrak{T}^{\pi} Z_1(s, a), \mathfrak{T}^{\pi} Z_2(s, a)) \\
&= D_{\mathrm{KL}}(R(s, a) + \gamma Z_1(s', a'), R(s, a) + \gamma Z_2(s', a')) \\
&= D_{\mathrm{KL}}(R(s, a) + Z_1(s', a'), Z_2(s', a')) \\
&\leq \sup_{s', a'} D_{\mathrm{KL}}(R(s, a) + Z_1(s', a'), Z_2(s', a')) \\
&= D_{\mathrm{KL}}^{\infty}(Z_1, Z_2)
\end{aligned}
\tag{23}
$$

There we have $D_{\mathrm{KL}}^{\infty}(\mathfrak{T}^{\pi} Z_1, \mathfrak{T}^{\pi} Z_2) \leq D_{\mathrm{KL}}^{\infty}(Z_1, Z_2)$, implying that $\mathfrak{T}^{\pi}$ is a non-expansive operator under $D_{\mathrm{KL}}^{\infty}$.

(2) By the definition of $D_{\mathrm{KL}}^{\infty}$, we have $\sup_{s, a} D_{\mathrm{KL}}(Z_n(s, a), Z(s, a)) \to 0$ implies $D_{\mathrm{KL}}(Z_n, Z) \to 0$. $D_{\mathrm{KL}}(Z_n, Z) \to 0$ implies the total variation distance $\delta(Z_n, Z) \to 0$ according to a straightforward application of Pinsker's inequality

$$
\begin{aligned}
\delta(Z_n, Z) &\leq \sqrt{\frac{1}{2} D_{\mathrm{KL}}(Z_n, Z)} \to 0 \\
\delta(Z, Z_n) &\leq \sqrt{\frac{1}{2} D_{\mathrm{KL}}(Z, Z_n)} \to 0
\end{aligned}
\tag{24}
$$

Based on Theorem 2 in WGAN (Arjovsky et al., 2017), $\delta(Z_n, Z) \to 0$ implies $W_p(Z_n, Z) \to 0$. This is trivial by recalling the fact that $\delta$ and $W$ give the strong an weak topologies on the dual of $(C(\mathcal{X}), \|\cdot\|_{\infty})$ when restricted to $\mathrm{Prob}(\mathcal{X})$.

(3) The conclusion holds because the $\mathfrak{T}^{\pi}$ degenerates to $\mathcal{T}^{\pi}$ regardless of the metric $d_p$. Specifically, due to the linearity of expectation, we obtain that

$$
\|\mathbb{E}\mathfrak{T}^{\pi} Z_1 - \mathbb{E}\mathfrak{T}^{\pi} Z_2\|_{\infty} = \|\mathcal{T}^{\pi} \mathbb{E} Z_1 - \mathcal{T}^{\pi} \mathbb{E} Z_2\|_{\infty} \leq \gamma \|\mathbb{E} Z_1 - \mathbb{E} Z_2\|_{\infty}.
\tag{25}
$$

This implies that the expectation of $Z$ under $D_{\mathrm{KL}}$ exponentially converges to the expectation of $Z^*$, i.e., $\gamma$-contraction. $\qquad\square$

## C  PROOF OF PROPOSITION 2

*Proof.* Firstly, given a fixed $p(x)$ we know that minimizing $D_{\mathrm{KL}}(p, q_\theta)$ is equivalent to minimizing $\mathcal{H}(p, q)$ by following

$$
\begin{aligned}
D_{\mathrm{KL}}(p, q_\theta) &= \int_{-\infty}^{+\infty} p(x) \log \frac{p(x)}{q_\theta(x)} \, \mathrm{d}x \\
&= -\int_{-\infty}^{+\infty} p(x) \log q_\theta(x) \, \mathrm{d}x - \left(-\int_{-\infty}^{+\infty} p(x) \log p(x) \, \mathrm{d}x\right) \\
&= \mathcal{H}(p, q_\theta) - \mathcal{H}(p) \\
&\propto \mathcal{H}(p, q_\theta)
\end{aligned}
\tag{26}
$$

Based on $\mathcal{H}(p, q_\theta)$, we can derive the objective function within each neural Z-fitted iteration as

$$\frac{1}{n} \sum_{i=1}^{n} \mathcal{H}(p^{s_i, a_i}(x), q_\theta^{s_i, a_i}(x))$$

$$= \frac{1}{n} \sum_{i=1}^{n} \left( -\int_{-\infty}^{+\infty} p^{s_i, a_i}(x) \log q_\theta^{s_i, a_i}(x) \, dx \right)$$

$$= \frac{1}{n} \sum_{i=1}^{n} \left( -\int_{-\infty}^{+\infty} \left( (1-\epsilon) \delta_{\{x = \mathbb{E}[Z^\pi(s, a)]\}}(x) + \epsilon \mu^{s_i, a_i}(x) \right) \log q_\theta^{s_i, a_i}(x) \, dx \right)$$

$$= \frac{1}{n} \sum_{i=1}^{n} \left[ (1-\epsilon) \left( -\int_{-\infty}^{+\infty} \delta_{\{x = \mathbb{E}[Z^\pi(s, a)]\}}(x) \log q_\theta^{s_i, a_i}(x) \, dx \right) + \epsilon \mathcal{H}(\mu^{s_i, a_i}, q_\theta^{s_i, a_i}) \right] \quad (27)$$

$$= \frac{1}{n} \sum_{i=1}^{n} \left[ (1-\epsilon) \mathcal{H}(\delta_{\{x = \mathbb{E}[Z^\pi(s_i, a_i)]\}}, q_\theta^{s_i, a_i}) + \epsilon \mathcal{H}(\mu^{s_i, a_i}, q_\theta^{s_i, a_i}) \right]$$

$$\propto \frac{1}{n} \sum_{i=1}^{n} \mathcal{H}(\delta_{\{x = \mathbb{E}[Z^\pi(s_i, a_i)]\}}, q_\theta^{s_i, a_i}) + \alpha \mathcal{H}(\mu^{s_i, a_i}, q_\theta^{s_i, a_i}), \text{ where } \alpha = \frac{\epsilon}{1-\epsilon} > 0$$

$$= \frac{1}{n} \sum_{i=1}^{n} \left( -\int_{-\infty}^{+\infty} \log q_\theta^{s_i, a_i}(\mathbb{E}[Z^\pi(s, a)]) \, dx + \alpha \mathcal{H}(\mu^{s_i, a_i}, q_\theta^{s_i, a_i}) \right)$$

$\square$

## D    PROOF OF THEOREM 1

*Proof.* Firstly, we have:

$$V(s_{t+1}) = \mathbb{E}_{a_{t+1} \sim \pi, x \sim \mu^{s_{t+1}, a_{t+1}}} \left[ Q(s_{t+1}, a_{t+1}) + \log q_\theta^{s_{t+1}, a_{t+1}}(x) \right]$$

$$= \mathbb{E}_{a_{t+1} \sim \pi} \left[ Q(s_{t+1}, a_{t+1}) \right] + \mathbb{E}_{a_{t+1} \sim \pi, x \sim \mu^{s_{t+1}, a_{t+1}}} \left[ \log q_\theta^{s_{t+1}, a_{t+1}}(x) \right]$$

$$= \mathbb{E}_{a_{t+1} \sim \pi} \left[ Q(s_{t+1}, a_{t+1}) \right] - \mathbb{E}_{a_{t+1} \sim \pi} \left[ -\int_{-\infty}^{\infty} \mu^{s_{t+1}, a_{t+1}}(x) \log q_\theta^{s_{t+1}, a_{t+1}}(x) \, dx \right]$$

$$= \mathbb{E}_{a_{t+1} \sim \pi} \left[ Q(s_{t+1}, a_{t+1}) \right] - \mathbb{E}_{a_{t+1} \sim \pi} \left[ \mathcal{H}\left( \mu^{s_{t+1}, a_{t+1}}, q_\theta^{s_{t+1}, a_{t+1}} \right) \right]$$

(28)

Further, we plug in $V(s_{t+1})$ into RHS of the iteration in Eq. 7, then we obtain

$$\mathcal{T}_d^\pi Q(s_t, a_t)$$
$$= r(s_t, a_t) + \gamma \mathbb{E}_{s_{t+1} \sim \rho^\pi} \left[ V(s_{t+1}) \right]$$
$$= r(s_t, a_t) - \gamma \mathbb{E}_{(s_{t+1}, a_{t+1}) \sim \rho_\pi} \left[ \mathcal{H}\left( \mu^{s_{t+1}, a_{t+1}}, q_\theta^{s_{t+1}, a_{t+1}} \right) \right] + \gamma \mathbb{E}_{(s_{t+1}, a_{t+1}) \sim \rho_\pi} \left[ Q(s_{t+1}, a_{t+1}) \right]$$
$$\triangleq r_\pi(s_t, a_t) + \gamma \mathbb{E}_{(s_{t+1}, a_{t+1}) \sim \rho_\pi} \left[ Q(s_{t+1}, a_{t+1}) \right],$$

(29)

where $r_\pi(s_t, a_t) \triangleq r(s_t, a_t) - \gamma \mathbb{E}_{(s_{t+1}, a_{t+1}) \sim \rho_\pi} \left[ \mathcal{H}\left( \mu^{s_{t+1}, a_{t+1}}, q_\theta^{s_{t+1}, a_{t+1}} \right) \right]$ is the entropy augmented reward we redefine. Thus, repeatedly updating via the above equations is equivalent to maximizing the new objective function $J'(\theta) = \sum_{t=0}^{T} \mathbb{E}_{(s_t, a_t) \sim \rho^\pi} \left[ r(s_t, a_t) - \gamma \mathcal{H}(\mu^{s_t, a_t}, q_\theta^{s_t, a_t}) \right]$. Applying the standard convergence results for policy evaluation (Sutton & Barto, 2018), we can attain that this Bellman updating under $\mathcal{T}_d^\pi$ is convergent under the assumption of $|\mathcal{A}| < \infty$ and bounded entropy augmented rewards $r_\pi$. $\square$

## E    DERIVATION OF HISTOGRAM DISTRIBUTIONAL LOSS

We show the derivation details of the Histogram distribution loss starting from KL divergence between $p$ and $q_\theta$. $p_i$ is the cumulative probability increment of target distribution $\{Y_i\}_{i \in [n]}$ within the $i$-th bin, and $q_\theta$ corresponds to a (normalized) histogram, and has density values $\frac{f_i^\theta(\mathbf{x}(s))}{w_i}$ per bin.

Thus, we have:

$$
\begin{aligned}
D_{\mathrm{KL}}\left(p^{s,a}, q_\theta^{s,a}\right) &= -\int_a^b p^{s,a}(y) \log q_\theta^{s,a}(y) dy \\
&= -\sum_{i=1}^k \int_{l_i}^{l_i+w_i} p^{s,a}(y) \log \frac{f_i^\theta(\mathbf{x}(s))}{w_i} dy \\
&= -\sum_{i=1}^k \log \frac{f_i^\theta(\mathbf{x}(s))}{w_i} \underbrace{\left(F^{s,a}\left(l_i + w_i\right) - F^{s,a}\left(l_i\right)\right)}_{p_i} \\
&\propto -\sum_{i=1}^k p_i^{s,a} \log f_i^\theta(\mathbf{x}(s))
\end{aligned}
\tag{30}
$$

where the last equality holds because the width parameter $w_i$ can be ignored for this minimization problem.

## F    PROOF OF LEMMA 1

*Proof.* For the histogram distributional loss below,

$$
\mathcal{L}_\theta(s,a) = -\sum_{i=1}^k p_i^{s,a} \log f_i^\theta(\mathbf{x}(s)), \text{ where } f_i^\theta(\mathbf{x}(s)) = \frac{\exp\left(\mathbf{x}(s)^\top \theta_i\right)}{\sum_{j=1}^k \exp\left(\mathbf{x}(s)^\top \theta_j\right)}
$$

we firstly prove its convexity. Note that $-\log \frac{\exp\left(\mathbf{x}(s)^\top \theta_i\right)}{\sum_{j=1}^k \exp(\mathbf{x}(s)^\top \theta_j)} = \log \sum_{j=1}^k \exp\left(\mathbf{x}(s)^\top \theta_j\right) - \mathbf{x}(s)^\top \theta_i$, the first term is Log-sum-exp, which is convex (see Convex optimization by Boyd and Vandenberghe), and the second term is affine function. Thus, $\mathcal{L}_\theta(s,a)$ is convex.

Secondly, we show that $\mathcal{L}_\theta(s,a)$ is $kl$-Lipschitz. We compute the gradient of the Histogram distributional loss regarding $\theta_i$:

$$
\begin{aligned}
&\frac{\partial}{\partial \theta_i} \sum_{j=1}^k p_j^{s,a} \log f_j^\theta(\mathbf{x}(s)) \\
&= \sum_{j=1}^k p_j^{s,a} \frac{1}{f_j^\theta(\mathbf{x}(s))} \nabla_{\theta_i} f_j^\theta(\mathbf{x}(s)) \\
&= \sum_{j=1}^k p_j^{s,a} \frac{1}{f_j^\theta(\mathbf{x}(s))} f_i^\theta(\mathbf{x}(s))(\delta_{ij} - f_j^\theta(\mathbf{x}(s)))\mathbf{x}(s) \\
&= \left(p_i^{s,a}(1 - f_i^\theta(\mathbf{x}(s))) - \sum_{j \neq i}^k p_j^{s,a} f_i^\theta(\mathbf{x}(s))\right) \mathbf{x}(s) \\
&= \left(p_i^{s,a} - p_i^{s,a} f_i^\theta(\mathbf{x}(s)) - (1 - p_i^{s,a}) f_i^\theta(\mathbf{x}(s))\right) \mathbf{x}(s) \\
&= \left(p_i^{s,a} - f_i^\theta(\mathbf{x}(s))\right) \mathbf{x}(s)
\end{aligned}
\tag{31}
$$

where $\delta_{ij} = 1$ if $i = j$, otherwise 0. Then, as we have $\|\mathbf{x}(s)\| \leq l$, we bound the norm of its gradient

$$
\begin{aligned}
&\|\frac{\partial}{\partial\theta}\sum_{j=1}^{k}p_j\log f_j^{\theta}(\mathbf{x}(s))\| \\
&\leq \sum_{i=1}^{k}\|\frac{\partial}{\partial\theta_i}\sum_{j=1}^{k}p_j\log f_j^{\theta}(\mathbf{x}(s))\| \\
&= \sum_{i=1}^{k}\|\left(p_i^{s;a} - f_i^{\theta}(\mathbf{x}(s))\right)\mathbf{x}(s)\| \\
&\leq \sum_{i=1}^{k}|p_i^{s,a} - f_i^{\theta}(\mathbf{x}(s))|\|\mathbf{x}(s)\| \\
&\leq kl
\end{aligned}
\tag{32}
$$

The last equality satisfies because $|p_i - f_i^{\theta}(\mathbf{x}(s))|$ is less than 1 and even smaller. Therefore, we obtain that $\mathcal{L}_\theta$ is $kl$-Lipschitz.

Lastly, we show that $\mathcal{L}_\theta$ is $kl^2$-Lipschitz smooth. A lemma is that $\log(1 + \exp(x))$ is $\frac{1}{4}$-smooth as its second-order gradient is bounded by $\frac{1}{4}$, and if $g(w)$ is $\beta$-smooth w.r.t. $w$, then $g(\langle x, w\rangle)$ is $\beta\|x\|^2$-smooth. Based on this knowledge, we firstly focus on the 1-dimensional case of function $\log f_j^{\theta}(z)$. As we have derived, we know that $\frac{\partial}{\partial\theta_i}\log f_j^{\theta}(z) = \delta_{ij} - f_i^{\theta}(z)$. Then the second-order gradient is $\frac{\partial^2\log f_j^{\theta}(z)}{\partial\theta_i\partial\theta_k} = -(\delta_{ik} - f_k^{\theta}(z)) = f_k^{\theta}(z) - 1$ if $i = k$, otherwise $f_k^{\theta}(z)$. Clearly, $|\frac{\partial^2\log f_j^{\theta}(z)}{\partial\theta_i\partial\theta_k}| \leq 1$, which implies that $\log f_j^{\theta}(z)$ is 1-smooth. Thus, $\log f_j^{\theta}(\langle x, \theta_i\rangle)$ is $\|x\|^2$-smooth, or $l^2$-smooth. Further, $\sum_{j=1}^{k}p_j^{s,a}\log f_j^{\theta}(\mathbf{x}(s))$ is also $l^2$-smooth as we have

$$
\begin{aligned}
&\|\nabla_{\theta_i}\sum_{j=1}^{k}p_j^{s,a}\log f_j^{\theta}(\mu) - \nabla_{\theta_i}\sum_{j=1}^{k}p_j^{s,a}\log f_j^{\theta}(\nu)\| \\
&\leq \sum_{j=1}^{k}p_j^{s,a}\|\nabla_{\theta_i}\log f_j^{\theta}(\mu) - \nabla_{\theta_i}\log f_j^{\theta}(\nu)\| \\
&\leq \sum_{j=1}^{k}p_j^{s,a}\cdot l^2\|\mu - \nu\| \\
&= l^2\|\mu - \nu\|
\end{aligned}
\tag{33}
$$

for each $\mu$ and $\nu$. Therefore, we further have

$$
\begin{aligned}
&\|\nabla_{\theta}\sum_{j=1}^{k}p_j^{s,a}\log f_j^{\theta}(\mu) - \nabla_{\theta}\sum_{j=1}^{k}p_j^{s,a}\log f_j^{\theta}(\nu)\| \\
&\leq \sum_{i=1}^{k}\|\nabla_{\theta_i}\sum_{j=1}^{k}p_j^{s,a}\log f_j^{\theta}(\mu) - \nabla_{\theta_i}\sum_{j=1}^{k}p_j^{s,a}\log f_j^{\theta}(\nu)\| \\
&\leq \sum_{i=1}^{k}l^2\|\mu - \nu\| \\
&= kl^2\|\mu - \nu\|
\end{aligned}
\tag{34}
$$

Finally, we conclude that $\mathcal{L}_\theta(s, a)$ is $kl^2$-smooth.

$\square$

## G  PROOF OF THEOREM 2

*Proof.* Consider the stochastic gradient descent rule as $G_{\lambda,\mathcal{L}}(\theta) = \theta - \lambda \nabla_\theta \mathcal{L}(\theta)$. Firstly, we provide two definitions about $\mathcal{L}_\theta$ for the following proof.

**Definition 3.** *($\sigma$-bounded) An update rule is $\sigma$-bounded if $\sup_\theta \|\theta - \lambda \nabla_\theta \mathcal{L}(\theta)\| \leq \sigma$.*

**Definition 4.** *($\eta$-expansive) An update rule is $\eta$-expansive if $\sup_{v,w} \frac{\|G_{\lambda,\mathcal{L}}(v) - G_{\lambda,\mathcal{L}}(w)\|}{\|u-w\|} \leq \eta$.*

**Lemma 3.** *(Grow Recursion, Lemma 2.5 (Hardt et al., 2016)) Fix an arbitrary sequence of updates $G_1, ..., G_T$ and another sequence $G'_1, ..., G'_T$. Let $\theta_0 = \theta'_0$ be the starting point and define $\delta_t = \|\theta'_i - \theta_t\|$, where $\theta_t$ and $\theta'_t$ are defined recursively through*

$$\theta_{t+1} = G_{\lambda,\mathcal{L}}(\theta_t), \ \theta'_{t+1} = G'_{\lambda,\mathcal{L}}(\theta'_t)$$

*Then we have the recurrence relation:*

$$\delta_{t+1} \leq \begin{cases} \eta \delta_t & G_t = G'_t \text{ is } \eta\text{-expansive} \\ \min(\eta, 1)\delta_t + 2\sigma_t & G_t \text{ and } G'_t \text{ are } \sigma\text{-bounded}, G_t \text{ is } \eta \text{ expansive} \end{cases}$$

**Lemma 4.** *(Lipschitz Continuity) Assume $\mathcal{L}_\theta$ is $L$-Lipschitz, the gradient update $G_{\lambda,\mathcal{L}}$ is $(\lambda L)$-bounded.*

*Proof.* $\|\theta - G_{\lambda,\mathcal{L}}(\theta)\| = \|\lambda \nabla_\theta \mathcal{L}(\theta)\| \leq \lambda L$ □

**Lemma 5.** *(Lipschitz Smoothness) Assume $\mathcal{L}_\theta$ is $\beta$-smooth, then for any $\lambda \leq \frac{2}{\beta}$, the gradient update $G_{\lambda,\mathcal{L}}$ is 1-expansive.*

*Proof.* Please refer to Lemma 3.7 in (Hardt et al., 2016) for the proof. □

Based on all the results above, we start to prove Theorem 2. Our proof is largely based on (Hardt et al., 2016), but it is applicable in distributional RL setting as well as considering desirable properties of histogram distributional loss. According Lemma 1, we attain that $\mathcal{L}_\theta$ is $kl$-Lipschitz as well as $kl^2$-smooth, and thus based on Lemma 4, $G_{\lambda,\mathcal{L}}$ is $(\lambda kl)$-bounded, and 1-expansive if $\lambda \leq \frac{2}{kl^2}$. In the step $t$, SGD selects samples that are both in $S$ and $S'$, with probability $1 - \frac{1}{n}$. In this case, $G_t = G'_t$, and thus $\delta_{t+1} \leq \delta_t$ as $G_t$ is 1-expansive based on Lemma 3. The other case is that samples selected are different with probability $\frac{1}{n}$, where $\delta_{t+1} \leq \delta_t + 2\lambda_t kl$ based on Lemma 3. Thus, if $\lambda_t \leq \frac{2}{kl^2}$ we have:

$$\mathbb{E}|\mathcal{L}(\theta_T; x) - \mathcal{L}(\theta'_T; x)| \leq kl\mathbb{E}[\delta_T], \text{ where } \delta_T = \|\theta_T - \theta'_T\|$$

$$\leq kl\left((1 - \frac{1}{n})\mathbb{E}[\delta_{T-1}] + \frac{1}{n}\mathbb{E}[\delta_{T-1}] + \frac{2\lambda_{T-1}kl}{n}\right)$$

$$= kl\left(\mathbb{E}[\delta_{T-1}] + \frac{2\lambda_{T-1}kl}{n}\right)$$

$$= kl\left(\mathbb{E}[\delta_0] + \sum_{t=0}^{T-1}\frac{2\lambda_t kl}{n}\right) \tag{35}$$

$$\leq \frac{2k^2l^2}{n}\sum_{t=0}^{T-1}\frac{2}{kl^2}$$

$$= \frac{4kT}{n}$$

Since this bounds hold for all $S$, $S'$ and $x$, we attain the uniform stability in Definition 1 for our histogram distributional loss applied in distributional RL.

Define the population risk as:

$$R[\theta] = \mathbb{E}_x \mathcal{L}(\theta; x)$$

and the empirical risk as:

$$R_S[\theta] = \frac{1}{n}\sum_{i=1}^{n}\mathcal{L}(\theta; x_i)$$

According to Theorem 2.2 in (Hardt et al., 2016), if an algorithm $\mathcal{M}$ is $\epsilon_{\text{stab}}$-uniformly stable, then the generalization gap is $\epsilon_{\text{stab}}$-bounded, i.e.,

$$|\mathbb{E}_{S,A}[R_S[\mathcal{M}(S)] - R[\mathcal{M}(S)]]| \le \epsilon_{\text{stab}}$$

$\square$

## H  PROOF OF LEMMA 2

$$\mathbb{E}_{(s,a)\sim\rho^\pi}\left[\|\nabla\mathcal{L}_\theta(p^{s,a}, q_\theta^{s,a})) - \nabla G(\theta)\|^2\right] \le (1-\epsilon)\sigma^2 + \epsilon\kappa\sigma^2. \tag{36}$$

*Proof.* As we know that $p^{s,a}(x) = (1-\epsilon)\delta_{\{x=\mathbb{E}[Z^\pi(s,a)]\}}(x) + \epsilon\mu^{s,a}(x)$, then we have:

$$\nabla\mathcal{L}_\theta(p^{s,a}, q_\theta^{s,a}) = (1-\epsilon)\nabla\mathcal{L}_\theta(\delta_{\{x=\mathbb{E}[Z^\pi(s,a)]\}}, q_\theta^{s,a}) + \epsilon\nabla\mathcal{L}_\theta(\mu^{s,a}, q_\theta^{s,a})$$

Therefore,

$$\mathbb{E}_{(s,a)\sim\rho^\pi}\left[\|\nabla\mathcal{L}_\theta(p^{s,a}, q_\theta^{s,a}))- \nabla G(\theta)\|^2\right]$$
$$\le \mathbb{E}_{(s,a)\sim\rho^\pi}\left[(1-\epsilon)\|\nabla\mathcal{L}_\theta(\delta_{\{x=\mathbb{E}[Z^\pi(s,a)]\}}, q_\theta^{s,a})) - \nabla G(\theta)\|^2 + \epsilon\|\nabla\mathcal{L}_\theta(\mu^{s,a}, q_\theta^{s,a})) - \nabla G(\theta)\|^2\right]$$
$$= (1-\epsilon)\sigma^2 + \epsilon\kappa\sigma^2,$$

$$\tag{37}$$

where the first inequality uses the triangle inequality of norm, i.e., $\|(1-\epsilon)\mathbf{a} + \epsilon\mathbf{b}\|^2 \le (1-\epsilon)\|\mathbf{a}\|^2 + \epsilon\|\mathbf{b}\|^2$, and the last equality uses the definition of the variance of $\mathcal{L}_\theta(\delta_{\{x=\mathbb{E}[Z^\pi(s,a)]\}}, q_\theta^{s,a})$ and $\mathcal{L}_\theta(\mu^{s,a}, q_\theta^{s,a})$. $\square$

## I  PROOF OF THEOREM 3

*Proof.* (1) If we only consider the expectation of $Z^\pi(s,a)$, the entropy-regularized MLE would degenerate to the pure MLE regarding $\delta_{\{x=\mathbb{E}[Z^\pi(s,a)]\}}$. As $\mathcal{L}_\theta(\delta_{\{x=\mathbb{E}[Z^\pi(s,a)]\}}, q_\theta^{s,a})$ is $kl^2$-smooth, we have

$$G(\theta_{t+1}) - G(\theta_t)$$
$$\le \langle\nabla G(\theta_t), \theta_{t+1} - \theta_t\rangle + \frac{kl^2}{2}\|\theta_{t+1} - \theta_t\|^2 \tag{38}$$
$$= -\lambda\left\langle\nabla G(\theta_t), \nabla\mathcal{L}_\theta(\delta_{\{x=\mathbb{E}[Z^\pi(s,a)]\}}, q_\theta^{s,a})\right\rangle + \frac{kl^2\lambda^2}{2}\|\nabla\mathcal{L}_\theta(\delta_{\{x=\mathbb{E}[Z^\pi(s,a)]\}}, q_\theta^{s,a})\|^2$$

where the last first equation is according to the definition of Lipschitz-smoothness, and the last second one is based on the updating rule of $\theta$. Next, we take the expectation on both sides,

$$\mathbb{E}[G(\theta_{t+1}) - G(\theta_t)]$$
$$\le -\lambda\mathbb{E}\left[\|\nabla G(\theta_t)\|^2\right] + \frac{kl^2\lambda^2}{2}\mathbb{E}\left[\|\nabla\mathcal{L}_\theta(\delta_{\{x=\mathbb{E}[Z^\pi(s,a)]\}}, q_\theta^{s,a}) - \nabla G(\theta_t) + \nabla G(\theta_t)\|^2\right]$$
$$\le -\lambda\mathbb{E}\left[\|\nabla G(\theta_t)\|^2\right] + \frac{kl^2\lambda^2}{2}\mathbb{E}\left[\|\nabla\mathcal{L}_\theta(\delta_{\{x=\mathbb{E}[Z^\pi(s,a)]\}}, q_\theta^{s,a}) - \nabla G(\theta_t)\|^2\right] + \frac{kl^2\lambda^2}{2}\mathbb{E}\left[\|\nabla G(\theta_t)\|^2\right]$$
$$= \frac{\lambda(kl^2\lambda - 2)}{2}\mathbb{E}\left[\|\nabla G(\theta_t)\|^2\right] + \frac{kl^2\lambda^2}{2}\sigma^2$$
$$\le -\frac{\lambda}{2}\mathbb{E}\left[\|\nabla G(\theta_t)\|^2\right] + \frac{kl^2\lambda^2}{2}\sigma^2$$

$$\tag{39}$$

where the first two equation hold because $\nabla G(\theta) = \mathbb{E}[\nabla\mathcal{L}_\theta]$ and the last inequality comes from $\lambda \le \frac{1}{kl^2}$. Through the summation, we obtain that

$$\mathbb{E}[G(\theta_T) - G(\theta_0)] \le -\frac{\lambda}{2}\sum_{t=0}^{T-1}\mathbb{E}\left[\|\nabla G(\theta_t)\|^2\right] + \frac{kl^2\lambda^2 T}{2}\sigma^2$$

We let $\mathbb{E}\left[G(\theta_T)\right] = 0$, we have

$$\frac{1}{T}\sum_{t=0}^{T-1}\mathbb{E}\left[\|\nabla G(\theta_t)\|^2\right] \leq \frac{2G(\theta_0)}{\lambda T} + kl^2\lambda\sigma^2$$

By setting $\lambda \leq \frac{\tau^2}{2kl^2\sigma}$ and $T = \frac{4G(\theta_0)}{\lambda\tau^2}$, we can have $\frac{1}{T}\sum_{t=0}^{T-1}\mathbb{E}\left[\|\nabla G(\theta_t)\|^2\right] \leq \tau^2$, implying that the degenerated MLE can achieve $\tau$-station point if the sample complexity $T = O(\frac{1}{\tau^4})$.

(2) and (3) We are still based on the $kl^2$-smoothness of $\mathcal{L}(p^{s,a}, q_\theta^{s,a})$.

$$
\begin{aligned}
&G(\theta_{t+1}) - G(\theta_t)\\
&\leq \langle \nabla G(\theta_t), \theta_{t+1} - \theta_t \rangle + \frac{kl^2}{2}\|\theta_{t+1} - \theta_t\|^2\\
&= -\lambda \langle \nabla G(\theta_t), \nabla\mathcal{L}_\theta(p^{s,a}, q_\theta^{s,a}) \rangle + \frac{kl^2\lambda^2}{2}\|\nabla\mathcal{L}_\theta(p^{s,a}, q_\theta^{s,a})\|^2\\
&= -\frac{\lambda}{2}\|\nabla G(\theta_t)\|^2 + \frac{\lambda}{2}\|\nabla G(\theta_t) - \nabla\mathcal{L}_\theta(p^{s,a}, q_\theta^{s,a})\|^2 + \frac{\lambda(kl^2\lambda - 1)}{2}\|\nabla\mathcal{L}_\theta(p^{s,a}, q_\theta^{s,a})\|^2\\
&\leq -\frac{\lambda}{2}\|\nabla G(\theta_t)\|^2 + \frac{\lambda}{2}\|\nabla G(\theta_t) - \nabla\mathcal{L}_\theta(p^{s,a}, q_\theta^{s,a})\|^2
\end{aligned}
\tag{40}
$$

where the second equation is based on $\langle \mathbf{a}, -\mathbf{b} \rangle = \frac{1}{2}\left(\|\mathbf{a} - \mathbf{b}\|^2 - \|\mathbf{a}\|^2 - \|\mathbf{b}\|^2\right)$, and the last inequality is according to $\lambda \leq \frac{1}{kl^2}$. After taking the expectation, we have

$$
\begin{aligned}
&\mathbb{E}\left[G(\theta_{t+1}) - G(\theta_t)\right]\\
&\leq -\frac{\lambda}{2}\mathbb{E}\left[\|\nabla G(\theta_t)\|^2\right] + \frac{\lambda}{2}\mathbb{E}\left[\|\nabla G(\theta_t) - \nabla\mathcal{L}_\theta(p^{s,a}, q_\theta^{s,a})\|^2\right]\\
&\leq -\frac{\lambda}{2}\mathbb{E}\left[\|\nabla G(\theta_t)\|^2\right] + \frac{\lambda}{2}\left((1 - \epsilon)\sigma^2 + \epsilon\kappa\sigma^2\right)
\end{aligned}
\tag{41}
$$

where the last inequality is based on Lemma 2. We take the summation, and therefore,

$$\mathbb{E}\left[G(\theta_T) - G(\theta_0)\right] \leq -\frac{\lambda}{2}\sum_{t=0}^{T-1}\mathbb{E}\left[\|\nabla G(\theta_t)\|^2\right] + \frac{T\lambda}{2}\left((1 - \epsilon)\sigma^2 + \epsilon\kappa\sigma^2\right)$$

We let $\mathbb{E}\left[G(\theta_T)\right] = 0$ and $\epsilon = \frac{1}{1+\kappa}$, then,

$$
\begin{aligned}
&\frac{1}{T}\sum_{t=0}^{T-1}\mathbb{E}\left[\|\nabla G(\theta_t)\|^2\right]\\
&\leq \frac{2G(\theta_0)}{\lambda T} + (1 - \epsilon)\sigma^2 + \epsilon\kappa\sigma^2\\
&= \frac{2G(\theta_0)}{\lambda T} + \frac{2\kappa}{1 + \kappa}\sigma^2\\
&\leq \frac{2G(\theta_0)}{\lambda T} + 2\kappa\sigma^2
\end{aligned}
\tag{42}
$$

If $\kappa \leq \frac{\tau^2}{4\sigma^2}$ and let $T = \frac{4G(\theta_0)}{\lambda\tau^2}$, this leads to $\frac{1}{T}\sum_{t=0}^{T-1}\mathbb{E}\left[\|\nabla G(\theta_t)\|^2\right] \leq \tau^2$, i.e., $\tau$-stationary point, with the sample complexity as $O(\frac{1}{\tau^2})$. Thus, (2) has been proved. On the other hand, if $\kappa > \frac{\tau^2}{4\sigma^2}$, we set $T = \frac{G(\theta_0)}{\lambda\kappa\sigma^2}$. This implies that $\frac{1}{T}\sum_{t=0}^{T-1}\mathbb{E}\left[\|\nabla G(\theta_t)\|^2\right] \leq 4\kappa\sigma^2 = O(\kappa)$. Therefore, the degree of stationary point is determined the degree of distribution approximation measured by $\kappa$. Thus, we obtain (3). $\qquad\square$

## J   PROOF OF THEOREM 4

*Proof.* 1. As $\varepsilon \to 0$, it is obvious to observe that Sinkhorn loss degenerates to the wasserstein distance.

2. As $\varepsilon \to \infty$, Sinkhorn loss turns to be $\int \log \left( \frac{\Pi(x,y)}{\mathrm{d}u(x)\mathrm{d}v(y)} \right) \mathrm{d}\Pi(x,y)$. We can consider its dual problem by introducing Lagrange multipliers $u$ and $v$. By additionally considering the primal-dual relation, we can solve the dual problem, which gives $u = v = 0$, and thus the optimal coupling is simply the product of the marginals , ie., $\Pi = u \otimes v$. Please refer to more detailed proof in Sinkhorn-GAN (Genevay et al., 2018). Since the cost function is the rectified kernel $k_\alpha$, under with MMD (Nguyen et al., 2020) can be $\gamma^{\alpha/2}$-contractive. $\qquad\square$

## K   REPRESENTATION EFFECT OF DISTRIBUTION RL

From the perspective of representation, we find that distributional RL encourages state representation from the same action class classified by the policy in tighter clusters.

The intrinsic characteristic of distributional RL is that it enables to learn a richer and more faithful representation of the environment, leading to more stable and efficient learning. To investigate the more informative representation resulting from distributional RL, we visualize how distributional RL changes the representation learned in the penultimate layer of the value network. We collect state features in the penultimate layer classified by the policy $\pi$ in different action classes, and perform t-SNE to reduce them to the two-dimensional space. In Figure 4, points in different colors represent state features classified into different action classes. It illustrates that both QR-DQN and C51 encourage the representation of state observations from the same action class to group in tighter clusters relative to expectation-based DQN. Interestingly, this phenomenon is similar to label smoothing (Müller et al., 2019), which leverages additional distributional knowledge in soft labels relative to hard labels. Therefore, we conclude that *similar to the benefit of label smoothing, distributional RL also encourages the activations of the penultimate layer to be close to the template of the correct action class and distant to the templates of the incorrect action classes.*

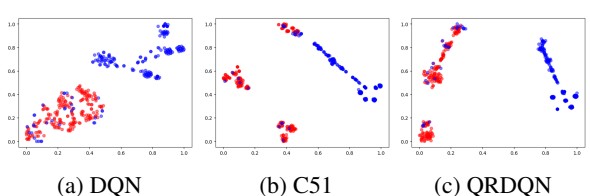

(a) DQN      (b) C51      (c) QRDQN

Figure 4: Visualization on the penultimate layer of the value network on Breakout. Each color denotes one action class.

## L   MORE EXPERIMENTAL RESULTS

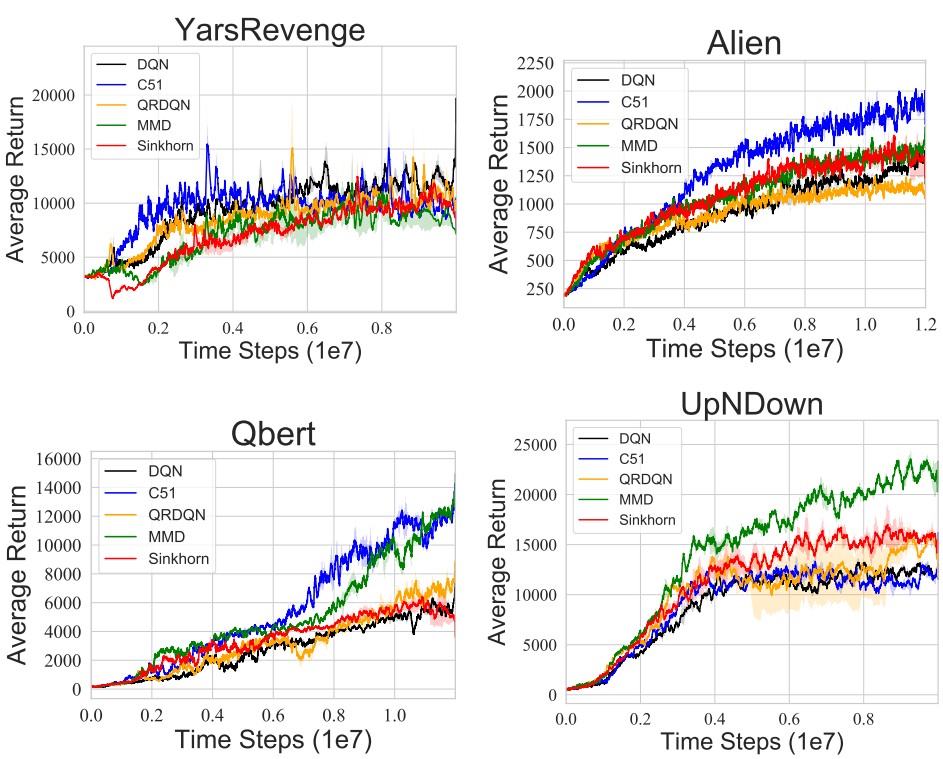

Figure 5: Performance of SinkhornDRL on YarRevenge, Aline, Qbert and UpNDown.

