# OpenReview forum: "Towards Understanding Distributional Reinforcement Learning: Regularization, Optimization, Acceleration and Sinkhorn Algorithm"
_ICLR.cc/2022/Conference — ICLR 2022 Submitted_

### Official Review · Reviewer_Qz4z · 2021-10-19

**Correctness:** 1
**Technical Novelty And Significance:** 1
**Empirical Novelty And Significance:** 1
**Recommendation:** 1
**Confidence:** 4

**Main Review:**

I don’t find any clear points or guarantees for distributional reinforcement learning. The authors provide some claims, but unfortunately, all of these claims do not directly relate to the benefits of distributional reinforcement learning.

Detailed Comments:

* On the regularization part:
  * Please note that, the decomposition of (4) is non-sense. I don’t find any interpretation of such results. Specifically why shall we separate the event $x = \mathbb{E}[Z^\pi(s, a)]$? After this decomposition, $F^{s, a}(x)$ is not the distribution of the value function any more, how should I parse the remaining results? Please also notice that when the value support on finite set that does not contain $\mathbb{E}[Z^\pi(s, a)]$, to make $F$ is still a valid probability measure, we can only let $\epsilon = 1$, which makes the remaining argument nonsense, as $\alpha = \frac{\epsilon}{1-\epsilon}$. Even the value support on a continuous set, after you make the claimed decomposition, the distribution significantly changes, as all of the events other than $x = \mathbb{E}[Z^\pi(s, a)]$ is down-scaled, which I don’t find any clear interpretation on that. Hence, I don’t think any of the claim for the regularization part makes any sense.
  * For Theorem 1, how can we know the exact $\mu(s^{t+1}, a^{t+1})$? If not, what’s the purpose of this theorem? Also, what’s the role of $\theta$ here? Are we aiming at the policy induced by $\theta$? Even the $Q$ sequence converge to the solution of that objective, what’s the theoretical property of the solution? When will it be used? What’s the theoretical benefit of using that? I have no idea on any of these questions.

* On the optimization and generalization part:
  * How do we partite the support of the state feature? Will it affect the accuracy? As each partition will lead to a totally different output probability, if I understand correctly?
  * The parameterization of $f$ doesn’t seem to partite the feature of $x$. It seems like perform a classification on the state, which means the authors partite the output rather than the input. Also, where are the actions take the role?
  * I don’t think properly match the density of each bin will provide a good estimation on the distribution, and hence I think if we minimize (8), we are not guaranteed to find the distribution of value.
  * I don’t check the proof of Theorem 2, but even it’s correct, it doesn’t mean anything. As I said, even we can perfectly match the probability on each bin, we don’t have any guarantees on the original distributional reinforcement learning problem. Can the authors state the guarantee of the original performance guarantee? For fitted Q-iteration, at least we can guarantee we can find the exact Q function under proper regularity condition. Also, the proof highly depends on the specific parameterization, I don’t think the authors can say such results can be held with neural network approximation. It’s never a ``neural’’algorithm.

* On the acceleration part:
  * As I said in the regularization part, we cannot set $\epsilon$, or otherwise we are not really doing distributional reinforcement learning.
  * Why should we parameterize $q$ in this way, if we only want the expectation? Theorem 3 (1) doesn’t make any sense to me.
  * As the authors don’t provide the bound for $\kappa$, and results of Theorem 3(3) is not satisfactory, I don’t get any information the authors want to argue.

* On the Sinkhorn part:
  * There are so many works for entropy regularized optimal transport, so I don’t think the loss is novel. Meanwhile, I don’t see any motivation on use this.

In summary, the authors never provide clear problem setup for each individual points and never provide rigorous theory on the benefits of distributional reinforcement learning compared with expectation based reinforcement learning. Instead, the authors only list several unrelated thing from other work and try to connect them with distributional reinforcement learning.

**Summary Of The Paper:**

The authors want to provide some new perspectives on distributional RL from the view of regularization, optimization, acceleration and eventually propose a class of Sinkhorn algorithm for distributional RL.

**Summary Of The Review:**

I don’t get any points the authors want to show, and the authors don’t provide any meaningful theoretical guarantees on the distributional reinforcement learning. The authors only list several unrelated thing and try to make them connected with distributional reinforcement learning, without going any real problems of distributional reinforcement learning.

---

> ### Author Response · Authors · 2021-11-21
> **Response**
>
> Thanks for your valuable comments. Below we address all your concerns. Please let us know if there are any further questions.
>
> **1.Regularization part.**
>
>  We respectfully disagree with you. Let us clarify.
>
> (1) The key motivation why we separate the full distribution into its expectation and remaining part is to decouple the advantage of distributional RL based on the remaining part over only the expectation-based RL from the whole value distribution.
>
> (2) We would like to clarify the correctness of Eq.4. Given the full distribution $F^{s, a}(x)$, when $x\le \mathbb{E(Z)}$ and the indicator function does not satisfy, $F^{s, a}(x)=\epsilon F_\mu(x)$ holds at the specific point $x$, which does not mean it always holds for any $x$. Thus, $\epsilon$ does not equal 1. Also, the definition of $F_\mu$ also satisfies the properties of a distribution function, including the 0 left limit, 1 right limit and the right continuity. More importantly, this decomposition is in fact common in the contamination model in robust statistics and label smoothing analysis. Please refer to more details from related works for its correctness.
>
> (3) For Theorem 1, $\mu(s^{t+1}, a^{t+1})$ is pre-specified for the analysis.  Also similar to the neural Z-fitted iteration, we update $q_\theta$ to approach $\mu$ based on the regularization effect, and therefore we indeed conduct the value iteration in Theorem 1 with convergence guarantee.
>
> **2.Optimization part.**
>
> (1) As we stated, we uniformly partition the support into k bins. The distribution histogram loss is mainly for the theoretical analysis, we do not leverage it in the design of real algorithms.
>
> (2) Not really. Eq.8 is the simplified result after the partition of support. Please refer to the detailed derivation in the appendix. In addition, $f^{s, a}(x)$ is the function of $s$ and $a$ and that is where the action takes the role.
>
> (3) In practice, it may be not, but we leverage it in the theoretical analysis. The comprehensive justification of this distribution histogram loss can refer to [1].
>
> (4) Please note that the distributional histogram loss is derived based on the KL divergence between the current and target value distribution within the neural Z-fitted iteration framework, which is well suitable for distributional RL. We also would like to clarify that we must find one way to characterize the distributional in order to allow the theoretical analysis in distributional RL. Thus, it is very natural to leverage a histogram to approximate the distribution.
>
>
> **3.Acceleration part.**
>
> (1) We only need the existence of $\epsilon$ to allow the theoretical analysis. This is well suitable for distributional RL.
>
> (2) As we clarified, we need to decouple the remaining part from the whole distribution getting rid of its expectation. Theorem 3(1) serves as a baseline of non-distributional RL and we thus can demonstrate distributional RL in Theorem 3(2) can converge faster than non-distributional RL.
>
> (3) Even though we do not bound $\kappa$ in Theorem (3) measuring the approximation error of distributional RL, we can still achieve a a reasonable performance, coinciding with previous empirical observations on various environments
>
> **4.Sinkhorn part.**
>
> To the best of our knowledge, Sinkhorn loss is the most commonly-used entropy regularized optimal transport method. Therefore, it is well motivated to introduce Sinkhorn loss to efficiently solve the Wasserstein distance in distributional RL.
>
> [1] Ehsan Imani and Martha White. Improving regression performance with distributional losses. (ICML 2018)

---

### Official Review · Reviewer_r4R2 · 2021-10-31

**Correctness:** 4
**Technical Novelty And Significance:** 2
**Empirical Novelty And Significance:** 2
**Recommendation:** 3
**Confidence:** 5

**Main Review:**

**Novelty**: The paper presents several interesting insights including the interpretation of distributional RL as entropy-regularized MLE and the finding of the representation of distributional RL that is different from that of expected RL. However, the other aspects including stability, optimization, generalization analysis, and Sinkhorn loss are quite straightforward applications from existing works. Details are as follows:
- The analysis in Section 4 is not significant nor relevant much to distributional RL. In particular, the paper considers a generalized linear model with softmax as the link function and analyzes the stability, optimization, and generalization in the supervised learning setting of such a generalized linear model with cross-entropy loss Eq. (8) which are quite a straightforward result from the literature. In addition, due to such simplification, their analysis's connection to distributional RL is weak.
- In RL, we often consider bounded reward, so if we consider the expected RL with squared loss function, it still enjoys all the properties of KL loss arising from distributional RL that is discussed in Section 4. Thus, the analysis in Section 4 does not show a significant advantage of distributional RL over expected RL.

**Clarity**: The paper is well structured though some paragraphs need to be improved for clarity. The paper studies several aspects of distributional RL (interpretation, optimization, generalization, representation, new algorithm) which are disconnected from each other. It might be clearer if the paper can focus on one aspect and go deeper analysis for that aspect instead of spreading out many disconnected ones. For example, with the proposed interpretation of distributional RL as entropy regularization, though being a nice interpretation, I am not sure what should we do with this fact to improve distributional RL? The paper can investigate deeply along this direction.

**Empirical significant**: The proposed Sinkhorn distributional RL performs quite on par with MMD without a clear-cut difference on the selected set of games. Moreover, the paper only presents on a small set of games. These make it hard to conclude whether there is empirical advantages in the proposed algorithm.

**Minor comments**:
- In Eq. (5), $\alpha$ should be explicitly written in terms of $\epsilon$
- The paragraph in Section 4 could be written better
- Eq (8) is not correct; it is an approximation with an extra entropy term
- Generalization guarantee is a consequence of uniform stability; thus Theorem 2 could be made less complicated in its statement.
- "our algorithm is not sensitive to the magnitude of $\epsilon$ as long as ..." in Section 8.2: not sure if I should interpret it as an advantage of the algorithm or a signal that something might have been wrong in your algorithm or implementation. It should be sensitive to $\epsilon$

**Summary Of The Paper:**

This paper interprets distributional RL as entropy-regularized MLE, explains distributional RL from the perspective of uniform stability, gives an insight into the representation of distributional RL as compared to that of expected RL, and proposes a Sinkhorn distributional RL algorithm that interpolates MMD and Wasserstein loss.

**Summary Of The Review:**

Though the paper proposes a nice interpretation of distributional RL as entropy regularization and the practical property of the representation of distributional RL, the other contributions seem do not significant or meaningful to me. However, the paper could be improved with the suggestions above as the idea presented in the paper is interesting and promising if it could be investigated more deeply. I feel the paper is not yet ready for publication at the current form, and I vote for rejection.

---

> ### Author Response · Authors · 2021-11-21
> **Response**
>
> We thank the reviewer's valuable remarks on our work. Below we address all your concerns. Please let us know if there are any further questions.
>
> **1.Novelty issues.**
>
> We respectfully disagree with your points. Firstly, distributional histogram loss is the result by solving a general KL divergence between the current and target distributions approximated by a histogram. In the neural Z-fitted iteration in Eq.3, if we leverage distributional histogram loss as the $d_p$, and it is right the case in the distributional RL. Distributional histogram loss is just similar to the mathematical form of the generalized linear model, but they are quite different.
>
> Secondly, we would like to clarify that the properties based on distribution histogram loss are derived by ourselves, without directly referring to other literature. We can also demonstrate that the least squared loss in expectation-based RL does not enjoy properties of RL despite the bounded rewards. In particular, we consider the gradient norm of least square loss regarding parameters $w$ in the linear function approximation case as
>
> $$| U_t-w_{t}^{\top} x_{t}| \Vert x_{t} \Vert$$
>
> where $U_t$ can be either an unbiased estimate via Monte Carlo method with $U_t=\sum_{k=0}^\infty \gamma^k r_{t+k+1}$, or a biased estimate via TD learning with $U_t=r_{t+1}+\gamma w_{t}^{\top} x_{t+1}$. Even though we have the bounded reward assumption, for example, $U_t  \in [\frac{R_{\text{min}}}{1-\gamma}, \frac{R_{\text{max}}}{1-\gamma}]$, we can not bound the $w_{t}^{\top} x_{t}$ and $\Vert x_t \Vert$ and thus it does not enjoy smooth properties in Lemma 1. By contrast, the histogram distributional loss in distribution RL can elegantly enjoy the smoothness and convexity properties as stated in Lemma 1.
>
> **2. Clarity.**
>
> We appreciate this suggestion. Both the algorithm part and representation section make the whole paper disconnected. As you suggested, we will focus on the regularization, optimization and acceleration sections with an empirical demonstration in the future version.
>
> **3.Empirical significance.**
>
> As we claimed, the performance of our Sinkhorn algorithm is competitive with SOTA methods, e.g., MMD. We would further improve our algorithm and implement it on all games in the future version.
>
>
> **4.Minor comments.**
>
> (1) $\alpha=\frac{\epsilon}{1-\epsilon}$, we present explicitly in the appendix.
>
> (2) $\epsilon$ in Sinkhorn indeed has an impact on the smoothness of loss if $\epsilon$ is overly small or large. However, we claim in our paper that our algorithm is
> not sensitive to the magnitude of $\epsilon$ as long as $\epsilon$ is within an appropriate interval, e.g., [1, 100].

---

### Official Review · Reviewer_Q4X1 · 2021-11-01

**Correctness:** 2
**Technical Novelty And Significance:** 2
**Empirical Novelty And Significance:** Not applicable
**Recommendation:** 3
**Confidence:** 3

**Main Review:**

Pros:
1. I think the general question this paper studies is important and also underexplored. A deeper understanding of the advantages of DRL will be beneficial to both methodological and applied research related to DRL. There have been few previous studies trying to formalize these advantages.
2. This paper proves some theoretical advantages of distributional RL from some new perspectives, such as optimization and sample efficiency. These, to my knowledge, are quite novel compared with previous work.


Concerns:
1. Section 4 and 5 show the theoretical advantages of distributional RL in terms of optimization stability, generalization, and sample efficiency. However, most results are based on one or more assumptions not used in popular distribution RL algorithms, such as assuming the optimizer as SGD with a certain learning rate, using the KL divergence as the distance measure. These assumptions are fine for theoretical analysis. However, to convince people that these theoretical results are significant, the paper should include some empirical evidence that these advantages do exist in practical distributional RL algorithms without these assumptions. Otherwise, it is not obvious whether these advantages come more from the distributional RL algorithm or these artificial assumptions.
2. The conclusion in Section 6, that distributional RL learns a better state representation, is weak and not well-supported. Figure 2 cannot be used to show that distributional RL does a better job in clustering representations than expectation-based RL. In fact, from humans’ eyes, all subfigures a, b, and c are quite similar in terms of clustering performance. Moreover, it is not sufficient to draw the strong conclusion at the end of Section 6 from just one visualization on one environment. More empirical and theoretical analysis are needed to make this claim.
3. Section 5 argues that distributional RL is more sample efficient than expectation RL. I find this result a bit counter-intuitive. Why does learning a distribution require fewer samples than its expectation? Also, Section 3 shows that distributional RL encourages exploration, which should lead to more samples required. Some intuitive clarifications here would help.
4. I cannot follow the proof of Theorem 4. It is difficult to see how the second case is related to MMD.
5. The main argument in Section 7, “Sinkhorn loss leverages the advantages of MMD and Wasserstein distance”, is not convincing. The paper shows two extreme cases (epsilon = 0, \infty) of Sinkhorn loss lead to MMD and Wasserstein distance. However, that does not automatically mean that the in-between cases (when epsilon is a positive number) will inherit the advantages of the two methods. In fact, they may get the disadvantages of MMD and Wasserstein distance. More empirical and theoretical analysis are needed to make this claim.
6. The paper only shows the results of 10 Atari games including the appendix. However, the Atari benchmark consists of 57 games, and each of them has very different environments and difficulties. The convention in the RL community is to do evaluations on all 57 Atari games, which is also the case for the previous work in distributional RL, such as, C51 and MMD. Also, there is no explanation for how these 10 games are selected from the benchmark. This makes these empirical results less trustworthy.
7. There seems to be a discrepancy between the results presented in the main paper (Figure 3) and the appendix (Figure 5). All games where the proposed method performs poorly are included in the appendix rather than the main paper. This selective presentation could mislead readers who haven’t read the appendix.

Minor suggestions
1. Overall, the paper can feel like some disconnected pieces glued together. The beginning of Section 7 says “Based on the theoretical analysis, we further design Sinkhorn distributional RL algorithm”. However, I could not see the connection between the proposed loss and the previous theoretical analysis. It would be good to point out these connections explicitly in writing, which will make the paper seem better-motivated and more pleasant to read.
2. I found some figures a bit confusing and found it difficult to know their purpose, such as Figure 1. I think it would help a bit to make the caption of each figure more detailed and self-contained.


**Summary Of The Paper:**

Distributional reinforcement learning (DRL) is a family of RL algorithms that estimates distributions of the value function rather than the expectation. The first part of the paper interprets distributional RL as adding a cross-entropy regularizer to the traditional objective. The second part discusses several advantages of DRL over expectation-based RL, including more stable optimization, better sample efficiency, and better state representations. The last part proposes a new distributional RL algorithm based on Sinkhorn loss, aiming to leverage the nice geometric interpretation of Wasserstein distance and sample complexity of Maximum Mean Discrepancy.

**Summary Of The Review:**

Overall, the questions that this paper wants to address are important, but the answers provided are not very satisfactory. The significance of most theoretical results in the paper is questionable due to the discrepancy with practical distributional RL algorithms (for example, most distributional RL do not use KL divergence to measure distance) and lack of empirical evidence to back up (such as Section 3, 4, 5). Some major claims are not well-supported by the rigorous evaluations (such as Section 6) and clear explanations (such as Section 4 and 5). The results section (Section 8) for the proposed method is not very trustworthy due to cherry-picking the results and lack of important details.

---

> ### Author Response · Authors · 2021-11-21
> **Response**
>
> We sincerely thank the reviewer's efforts on the valuable comments. Below we address all your concerns. Please let us know if there are any further questions.
>
> **1.Demonstration of Sections 4 and 5.**
>
>  We really appreciate this suggestion, as we also feel additional demonstration regarding Section 4 and 5 would make our paper more convincing as just like you mentioned the assumptions based on KL might not be applicable in real applications. Due to the limited time in rebuttal, we will add the empirical section as you suggested in the future version.
>
>
> **2.Section 6**
>
>  We agree with this suggestion. For the comprehensiveness, we introduce the representation part, but it seems to be disconnected. We thus defer this section into the appendix in the revised version.
>
> **3.Sample efficiency of distributional RL.**
>
> We would like to clarify that distributional RL has a more informative representation——it represents the value distribution in the final output rather than the value expectation. Learning a distributional may not necessarily need more samples. In addition, we also need to clarify distributional RL encourages exploration only when the true state-action distribution has a higher degree of dispersion, e.g., a larger variance, as stated in the last paragraph in Section 3. Even when distributional RL encourages exploration, it may sacrifice in the early phase, but it can still increase sample efficiency during the whole training phase due to the optimization and acceleration benefits.
>
> **4.Proof of Theorem 4.**
>
>  We will update the detailed proof of Theorem 4 in the future version.
>
> **5.The main argument of Section 7.**
>
>  In the current version, the connection between Sinkhorn and MMD/Wasserstein has been established only in the limiting situation. For the general case, the entropic regularization in the Sinhorn loss restricts the search space on the optimal transport that eventually corresponds to a Gibbs kernel, which can also be viewed as a variant of moment matching. We will add more explanation in the future version.
>
> **6.10 Atari games.**
>
> We select 10 typical and common games and reported results in the paper, but we would perform our algorithm on all 57 games in the future version as you suggested.
>
> **7.The discrepancy between Figure 3 in the main paper and Figure 5 in the appendix.**
>
> We only have a safe claim that our Sinkhorn algorithm is competitive to other SOTA distributional RL algorithms, e.g., MDD. Thus, it might be the case where the Sinhorn algorithm performs slightly worse than other SOTA algorithms on certain games. We will further improve our algorithm in the future version.

---

> > ### Comment · Reviewer_Q4X1 · 2021-11-25
> > **score unchanged**
> >
> > I would like to thank the author for the response. I'm happy with the explanation for concern 3. However, other concerns are not fully addressed in the current version. Thus, I will keep my score unchanged.

---

### Official Review · Reviewer_GhjJ · 2021-11-04

**Correctness:** 4
**Technical Novelty And Significance:** 2
**Empirical Novelty And Significance:** 2
**Recommendation:** 5
**Confidence:** 4

**Main Review:**

Generally speaking, the paper is poorly written; many parts were hard to follow, but more importantly, the different sections feel very disconnected. For example:

- Eq. 4 is poorly explained; the term F_\mu is not formally defined
- Section 3.3 was hard to follow; it wasn’t clear exactly what connection is being drawn
- Section 6 is very disconnected; there is no theory here but a small experiment, and the connection to the other results is unclear.
- Section 7 claims that the Sinkhorn approach is motivated by the theoretical analysis in the previous sections, but I don’t see why this is the case.

The paper also does not give sufficient background for the Sinkhorn loss

The paper currently feels like a disjoint collection of results, none of which are properly motivated or connected to other results. I believe the authors should focus on one or two key results and describe them and their importance in more detail, rather than try to present so many results without sufficient explanation.

Along the same lines, the significance of the results is unclear. For example, consider Theorem 2 in Section 4. How does this result differ from a corresponding result in the non-distributional setting (or to a different algorithm)? It would be significantly more helpful if such a contrast is drawn, since it would help clarify what is special about distributional reinforcement learning compared to traditional reinforcement learning. A similar comment goes for Theorem 3 in Section 5; this theorem had a bit more discussion, but it was hard to follow due to the conciseness.

Finally, the use of KL divergence in the objective is non-standard; a more typical approach (as the authors mention) is to use the Wasserstein or Cramer distances. Do the authors expect their insights to generalize to other objectives? A lot of their results appear to hinge pretty strongly on this formulation (e.g., the connection to maximum entropy RL).


**Summary Of The Paper:**

This paper performs a theoretical analysis of distributional RL. Furthermore, they propose a new algorithm based on the Sinkhorn loss, and empirically demonstrate that their algorithm outperforms existing approaches.


**Summary Of The Review:**

Pros
- Important problem

Cons
- Poor writing
- Disjointed and poorly motivated results
- Non-standard formulation

---

> ### Author Response · Authors · 2021-11-21
> **Response**
>
> We thank the reviewer's valuable remarks on our work. Below we address all your concerns. Please let us know if there are any further questions.
>
> **1. Writing issue.**
>
> (1) As stated before Eq.4,  we denote the distribution on the remaining support getting rid of $\mathbb{E}\left[Z^\pi\right]$ as $F_\mu$.
>
> (2) We hope to elaborate the connection between the derived cross-entropy regularization in Eq.5 with the classical entropy regularization in the maximum entropy RL, in which we further characterize their difference and interpret ours as a risk-aware regularization.
>
> (3) For comprehensiveness, we introduce Section 6 in the main body of our paper, but you are right as Section 6 seems to be disconnected from other parts, and we have put it in the appendix in the revised version.
>
> (4) We agree with you, and we split the theoretical analysis and new algorithm introduction into two parts. We have revised the conjunction sentences in the revised version.
>
> **2. Discussion of Sinkhorn.**
>
> Sinkhorn loss is a typical entropy regularization-based Wasserstein loss that makes the computation of Wasserstein distance attainable. We have tried our best to give a detailed definition of Sinkhorn loss in Eq.12 and 13, but it seems you are right and these descriptions may not be sufficient enough. Thus, we will provide a more detailed background in the future version.
>
> **3. Disconnected issues.**
>
> We would like to clarify that the regularization part is relatively disconnected from the optimization and acceleration parts. However, the acceleration part can be viewed as the extension of the optimization section as in the acceleration section we further decompose $F^{s, a}(x)$ via Eq.4 based on the distributional loss mentioned in the optimization section. We will polish our writing and clarify them more clearly in the future version.
>
> **4. Significance of results.**
>
> Due the the smoothness and convexity properties as stated in Lemma1, which is induced from the distributional loss in distributional RL, we can derive the stable optimization and acceleration effect (Theorems 2 and 3). By contrast, in the non-distributional RL case, the least-squared loss cannot guarantee the smoothness and convexity properties, and thus we cannot derive these results directly as least based on our analysis. We will provide a more detailed comparison in the future version.
>
> **5. Non-standard formulation and extension to Wasserstein distance.**
>
> The leverage of KL divergence facilitates the theoretical analysis. If we only pay attention to the Wasserstein distance, we may not obtain any insights into the theoretical advantages of distributional RL. At least, we can say our analysis based on KL is the first step, and we expect our work can inspire the extension work based on Wasserstein distance in the future.

---

### Decision · Program_Chairs · 2022-01-20

**Decision:**

Reject

**Comment:**

It appears that the reviewers have reached a consensus that the paper is not ready for publication at ICLR.